# Non-Markovian Discrete Diffusion with Causal Language Models

**Yangtian Zhang**[*]  **Sizhuang He**[*]  **Daniel Levine**  **Lawrence Zhao**  **David Zhang**
**Syed Asad Rizvi**  **Shiyang Zhang**  **Emanuele Zappala**  **Rex Ying**[†]  **David van Dijk**[†]
Yale University, New Haven, CT, USA
{yangtian.zhang, sizhuang.he, daniel.levine, lawrence.zhao}@yale.edu
{david.zhang, syed.rizvi, shiyang.zhang}@yale.edu   emanuele.zappala@isu.edu
{rex.ying, david.vandijk}@yale.edu

## Abstract

Discrete diffusion models offer a flexible, controllable approach to structured sequence generation, yet they still lag behind causal language models in expressive power. A key limitation lies in their reliance on the Markovian assumption, which restricts each step to condition only on the current state, leading to potential uncorrectable error accumulation. In this paper, we introduce **CaDDi** (Causal Discrete Diffusion Model), a discrete diffusion model that conditions on the entire generative trajectory, thereby lifting the Markov constraint and allowing the model to revisit and improve past states. By unifying sequential (causal) and temporal (diffusion) reasoning in a single non-Markovian transformer, CaDDi also treats standard causal language models as a special case and permits the direct reuse of pretrained LLM weights with no architectural changes. Empirically, CaDDi outperforms state-of-the-art discrete diffusion baselines on natural-language benchmarks, substantially narrowing the remaining gap to large autoregressive transformers.

## 1   Introduction

Autoregressive transformers have become a dominant approach for sequence modeling [60, 12, 56], achieving state-of-the-art performance in many natural language and biological tasks. Their left-to-right decoding paradigm simplifies training via next-token prediction and is supported by large-scale pretraining, unlocking broad linguistic (or domain) knowledge. However, these models can be less flexible for bidirectional or partially specified generation, such as text infilling or prompting from arbitrary locations.

By contrast, discrete diffusion models [17, 2, 24, 21] naturally accommodate controllable generation scenarios where tokens can be iteratively refined and sampled in a bidirectional manner [47]. Recent advances have extended discrete diffusion to continuous time [5, 48], improved its training objectives [48, 37, 44, 46], and accelerated inference [41, 36]. Yet, these models often lag behind autoregressive approaches in generation quality [66], due in part to their reliance on a *single* latent state for denoising—leading to fragile inference where small decoding errors can accumulate over time [63, 66, 32].

To address this limitation, in this work, we explore a non-Markovian diffusion framework to discrete sequences by allowing each denoising step to condition on the entire generative trajectory, rather than a single latent state. While prior work such as DART [23] has investigated non-Markovian diffusion for continuous data, its formulation relies on continuous kernels and smooth transitions that do not directly

---

[*]Equal contribution

[†]Correspondence to: rex.ying@yale.edu, david.vandijk@yale.edu

translate to the discrete setting. Notably, when instantiated for discrete sequences, non-Markovian diffusion naturally mirrors the structure of token-level autoregressive models—highlighting a close connection between the two.

Leveraging this insight, we propose CaDDi, a causal discrete diffusion model that unifies *sequential* (left-to-right) and *temporal* (multi-step) dimensions in a single decoder-only transformer architecture. As a result, CaDDi can be trained efficiently via a simple next-block/next-token prediction loss—similar to a causal language model—while preserving the bidirectional control and iterative refinement of diffusion.

Additionally, we show that its token-level variation CaDDi-AR can be viewed as a generalization of traditional autoregressive models ($T = 1$ is the special case), making it straightforward to fine-tune a pretrained LLM for discrete diffusion. Such adaptation unlocks flexible generation modes (e.g., text infilling) without sacrificing the rich knowledge encoded by large-scale pretraining.

In summary, our key contributions are as follows.

- We introduce a non-Markovian discrete diffusion framework where each denoising step incorporates the full generative trajectory, improving inference robustness.

- We propose CaDDi, a causal discrete diffusion model that unifies sequential and temporal modeling within a non-Markovian diffusion framework. Its further variation CaDDi-AR generalizes traditional causal language models as a special case and can seamlessly adopt pretrained LLMs for discrete diffusion, enabling more controllable and structured generation.

- Quantitative results show that CaDDi outperforms recent discrete diffusion models, achieving lower generative perplexity on language datasets and stronger reasoning capabilities when leveraging a pretrained LLM.

## 2   General Discrete Diffusion

Recently, discrete diffusion models [2] have emerged as a powerful framework. In contrast to their continuous counterparts, which corrupt data by adding Gaussian noise in a real-valued space, discrete diffusion models operate on categorical variables, gradually corrupting tokens before reconstructing them through a learned denoising process.

**Forward Process.**   Let $\mathbf{x}_0 = (\mathbf{x}_0^1, \mathbf{x}_0^2, \ldots, \mathbf{x}_0^L)$ be a sequence of discrete tokens from a vocabulary $\mathcal{V}$ of size $|\mathcal{V}|$. The forward (noising) process produces latent variables $\mathbf{x}_1, \mathbf{x}_2, \ldots, \mathbf{x}_T$ by iteratively corrupting each token according to a time-dependent transition matrix $\mathbf{Q}_t$:

$$q\big(\mathbf{x}_t \mid \mathbf{x}_{t-1}\big) = \mathrm{Cat}\Big(\mathbf{x}_t;\ \mathbf{x}_{t-1}\,\mathbf{Q}_t\Big), \tag{1}$$

where $\mathrm{Cat}(\cdot;\ \pi)$ denotes the categorical distribution with parameter $\pi$, and $[\mathbf{Q}_t]_{ij}$ gives the probability of transitioning from state $i$ to state $j$ at time $t$. Due to the Markovian assumption, the marginal distribution at any timestep can be computed in closed form:

$$q(\mathbf{x}_t \mid \mathbf{x}_0) = \mathrm{Cat}(\mathbf{x}_t; \mathbf{x}_0 \bar{\mathbf{Q}}_t), \quad \text{where} \quad \bar{\mathbf{Q}}_t = \mathbf{Q}_1 \mathbf{Q}_2 \ldots \mathbf{Q}_t. \tag{2}$$

Common choices for the transition kernel $\mathbf{Q}_t$ include uniform kernel $\mathbf{Q_t} = (1 - \beta_t)\mathbf{I} + \beta_t \mathbf{1}\mathbf{1}^T/|\mathcal{V}|$ of size $|\mathcal{V}| \times |\mathcal{V}|$ and absorbing kernel $\mathbf{Q}_t = (1 - \beta_t)\,\mathbf{I} + \beta_t \mathbf{1}e_m^\top$ of size $(|\mathcal{V}| + 1) \times (|\mathcal{V}| + 1)$, where $\mathbf{1}$ is an all-one vector and $e_m^\top$ represents a one-hot vector where element at index $m = |\mathcal{V}| + 1$ is 1.

**Reverse Process.**   The reverse (denoising) model $p_\theta\big(\mathbf{x}_{t-1} \mid \mathbf{x}_t\big)$ is commonly parameterized in form of posterior distribution , which is learned to reverse the corruption process

$$p_\theta\big(\mathbf{x}_{t-1} \mid \mathbf{x}_t\big) = q\big(\mathbf{x}_{t-1} \mid \mathbf{x}_t, \mathbf{x}_0 = \boldsymbol{\mu}_\theta\left(\mathbf{x}_t, t\right)\big) = \mathrm{Cat}\left(\mathbf{x}_{t-1}; \frac{\mathbf{x}_t \mathbf{Q}_t^\top \odot \boldsymbol{\mu}_\theta\left(\mathbf{x}_t, t\right) \bar{\mathbf{Q}}_{t-1}}{\boldsymbol{\mu}_\theta\left(\mathbf{x}_t, t\right) \bar{\mathbf{Q}}_t \mathbf{x}_t^\top}\right) \tag{3}$$

where $\boldsymbol{\mu}_\theta\left(\mathbf{x}_t, t\right) = p_\theta(\mathbf{x}_0 \mid \mathbf{x}_t)$ is a time-dependent denoiser (*$x_0$-parameterization*) mapping $\mathbf{x}_t$ back to the mean distribution of clean data.

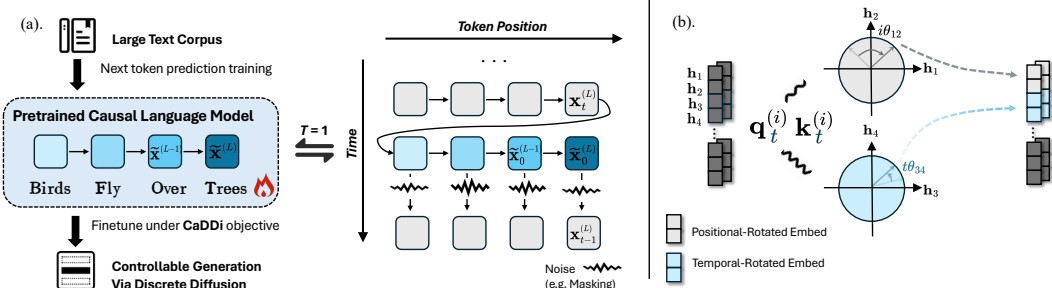

Figure 1: (a). **Inference paradigm for a standard causal language model versus CaDDi-AR.** In CaDDi-AR, each timestep first autoregressively denoises the tokens into $\widetilde{\mathbf{x}}_0$, then re-applies noise via the diffusion kernel to obtain $\mathbf{x}_{t-1}$. A traditional autoregressive model emerges as the special case of $T = 1$, which can be adapted to discrete diffusion by fine-tuning. (b). **Extending 1D to 2D Rotary Positional Encoding.** Standard rotary encodings for token positions are seamlessly generalized to also encode diffusion timesteps, remaining fully backward-compatible with existing language model architectures.

**Error During Inference.** During inference, discrete diffusion models can potentially encounter a variety of sources of error, including inaccuracies arising from heavy-tailed per-token sampling distributions, and independent factorization in denoising distribution $p_\theta\left(\mathbf{x}_0 \mid \mathbf{x}_t\right) = \Pi_i p_\theta\left(\mathbf{x}_0^i \mid \mathbf{x}_t\right) = \Pi_i \boldsymbol{\mu}_\theta^i\left(\mathbf{x}_t, t\right)$, which prevents the model from fully capturing the true posterior $q\left(\mathbf{x}_0 \mid \mathbf{x}_t\right)$ that inherently exhibits dependencies across token dimensions [63].

Although each individual source of error may be moderate, they can propagate and accumulate across reverse steps. Specifically, incorrectly predicted $\mathbf{x}_t$ will not only impact the immediate denoiser output $\boldsymbol{\mu}_\theta\left(\mathbf{x}_t, t\right)$, but also affect subsequent predictions, leading to further deviations as the trajectory progresses. Even when denoiser predictions are accurate, the constrained form of the approximated posterior can limit the model's ability to self-correct earlier mistakes.[3]

## 3 Non-Markovian Discrete Diffusion

### 3.1 Is the Markovian Assumption Necessary?

Diffusion models can be viewed as a special class of hierarchical variational autoencoders (HVAEs) with a Markovian assumption, where the latent variables consist of progressively corrupted versions of the original data. Let $\mathbf{x}_0$ be the original data and $\mathbf{x}_{1:T}$ be the latent variables at timesteps $1, \ldots, T$. The ELBO objective of HVAE can be written as:

$$\max_{\theta, \phi} \quad \mathcal{L}_{\theta, \phi}^{\text{ELBO}} = \mathbb{E}_{\mathbf{x}_{1:T} \sim q_\phi(\mathbf{x}_{1:T} | \mathbf{x}_0)}\left[\sum_{t=1}^{T} \log p_\theta\left(\mathbf{x}_{t-1} \mid \mathbf{x}_{t:T}\right) + \log p_\theta\left(\mathbf{x}_T\right) - \log q_\phi\left(\mathbf{x}_{1:T} \mid \mathbf{x}_0\right)\right],$$

(4)

where $q_\phi(\mathbf{x}_{1:T} \mid \mathbf{x}_0)$ is the variational posterior distribution, and $p_\theta(\mathbf{x}_{t-1} \mid \mathbf{x}_{t:T})$ is the generative (reverse) model. Most of the existing discrete diffusion models make Markovian assumptions by specifying a fixed, non-learnable forward process $q_\phi\left(\mathbf{x}_{1:T} \mid \mathbf{x}_0\right) = \prod_{t=1}^{T} q\left(\mathbf{x}_t \mid \mathbf{x}_{t-1}\right)$, and modeling the reverse process as a time-localized transition $p_\theta\left(\mathbf{x}_{t-1} \mid \mathbf{x}_{t:T}\right) = p_\theta\left(\mathbf{x}_{t-1} \mid \mathbf{x}_t\right)$, which is approximated by a posterior estimator such as $q\left(\mathbf{x}_{t-1} \mid \mathbf{x}_t, \mathbf{x}_0 = \mu_\theta(\mathbf{x}_t, t)\right)$. While beneficial for computational efficiency, the Markovian assumption compresses all relevant information into a single state $\mathbf{x}_t$, making the model more susceptible to error accumulation through the generative trajectory, which is shown in prior work [63] and our experiments in Figure 4.

In this work, we explore a more flexible alternative: treating discrete diffusion models as general HVAEs *without requiring the Markovian assumption*. Inspired by recent studies [23], we consider a non-Markovian generative process, where the reverse model $p_\theta\left(\mathbf{x}_{t-1} \mid \mathbf{x}_{t:T}\right)$ can access the

---

[3]One illustrative case of such limited self-correction is the *failure-to-remask* phenomenon under absorbing diffusion kernels, See Appendix

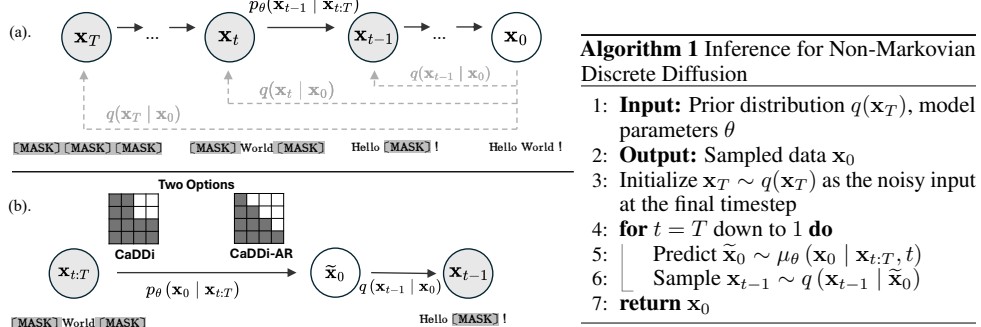

Figure 2: Illustration of the Non-Markovian discrete diffusion inference algorithm (right) and its forward/reverse process visualization (left).

entire future trajectory $\mathbf{x}_{t:T}$ to denoise $\mathbf{x}_{t-1}$ for denoising in an autoregressive manner. Assuming an independently parameterized forward corruption process, the reverse model simplifies to: $p_\theta\left(\mathbf{x}_{t-1} \mid \mathbf{x}_{t:T}\right) = q\left(\mathbf{x}_{t-1} \mid \mathbf{x}_0 = \mu_\theta\left(\mathbf{x}_{t:T}, t\right)\right)$, relaxing the structural constraints imposed by the Markovian factorization in both the forward and reverse directions.

In the following, we will describe how the non-Markovian discrete diffusion process is constructed. Crucially, we will see that the resulting non-Markovian autoregressive inference mechanism essentially aligns with a causal language model plus an additional temporal dimension—laying the groundwork for our unified spatial-temporal framework (Section 4).

## 3.2 Non-Markovian Forward Process

Following [23], we adopt a simple yet expressive *independent* noising process to ensure that the latent trajectory carry more complementary information across timesteps. Instead of relying on a stepwise Markov chain, we inject independent noise into the original data $\mathbf{x}_0$ at each timestep, as illustrated in Fig. 2. Formally, the forward process is defined as:

$$q(\mathbf{x}_{0:T}) := q(\mathbf{x}_0, \mathbf{x}_1, \mathbf{x}_2, ..., \mathbf{x}_T) = q(\mathbf{x}_0) \prod_{t=1}^{T} q(\mathbf{x}_t | \mathbf{x}_{0:t-1}) = q(\mathbf{x}_0) \prod_{t=1}^{T} q(\mathbf{x}_t | \mathbf{x}_0), \tag{5}$$

where the final equality follows from our assumption that noise is added independently at each timestep, conditioned only on $\mathbf{x}_0$. While the per-step marginals $q\left(\mathbf{x}_t \mid \mathbf{x}_{0:t-1}\right) = q\left(\mathbf{x}_t \mid \mathbf{x}_0\right)$ appear similar to those in Markovian diffusion models, the conditional independence of $\mathbf{x}_t$ from $\mathbf{x}_{t-1}$ (given $\mathbf{x}_0$) introduces a fundamentally different structure into the forward trajectory.

In this non-Markovian formulation, the forward process depends solely on the marginal corruption kernel $q\left(\mathbf{x}_t \mid \mathbf{x}_0\right)$, rather than the usual Markovian kernel $q\left(\mathbf{x}_t \mid \mathbf{x}_{t-1}\right)$. In practice, one can still leverage standard discrete diffusion kernels-such as the absorbing or uniform kernels-or even mixtures thereof, without enforcing a temporal Markov constraint. Additionally, we can establish the following correspondence:

**Proposition 3.1.** *An absorbing-state non-Markovian discrete diffusion process with marginal transition kernel $\bar{\mathbf{Q}}_t = (1 - \alpha_t)\,\mathbf{I} + \alpha_t \mathbf{1} e_m^\top$ admits a bijection to an absorbing-state Markovian discrete diffusion process with marginal transition kernel $\bar{\mathbf{Q}}_t^* = (1 - \alpha_t^*)\,\mathbf{I} + \alpha_t^* \mathbf{1} e_m^\top$ such that the two processes exhibit identical mutual information decay between $x_{t:T}$ and $x_0$, provided that the coefficients satisfy $\alpha_t^* = \prod_{\tau=t}^{T} \alpha_\tau$. Proofs is provided in Appendix A.1*

## 3.3 Non-Markovian Inference Process

Similar to conventional diffusion models, our non-Markovian discrete diffusion model employs a posterior-inspired parameterization for the reverse process $p_\theta\left(\mathbf{x}_{t-1} \mid \mathbf{x}_t\right)$ However, due to the assumption of an independent corruption kernel—where the forward noise depends only on $\mathbf{x}_0$—we can further simplify the posterior form:

$$p_\theta(\mathbf{x}_{t-1} \mid \mathbf{x}_{t:T}) := q\big(\mathbf{x}_{t-1} \mid \mathbf{x}_{t:T}, \mathbf{x}_0 = \mu_\theta\left(\mathbf{x}_{t:T}, t\right)\big) = q\big(\mathbf{x}_{t-1} \mid \mathbf{x}_0 = \mu_\theta\left(\mathbf{x}_{t:T}, t\right)\big) \tag{6}$$

Notice that it lift the constraint introduced by $\mathbf{x}_{t:T}$ in the posterior form and delegating all the flexibility for the denoiser model $\mu_\theta$. As illustrated in Fig. 2 (right), inference proceeds in an autoregressive manner: at each timestep, we first predict a clean estimate $\widetilde{\mathbf{x}}_0 \sim \mu_\theta(\mathbf{x}_{t:T}, t)$, then sample $\mathbf{x}_{t-1}$ using the forward corruption kernel $q(\mathbf{x}_{t-1} \mid \widetilde{\mathbf{x}}_0)$. This loop continues backward through time until the final data sample $\mathbf{x}_0$ is recovered.

## 3.4 Evidence Lower Bound

As with traditional discrete diffusion models, the non-Markovian variant can be trained using a variational objective. Specifically, we optimize the Evidence Lower Bound (ELBO) as:

$$\mathcal{L}_{\text{non-markov}} = \mathbb{E}_{q(\mathbf{x}_{1:T}|\mathbf{x}_0)} \, \log p_\theta(\mathbf{x}_0 \mid \mathbf{x}_{1:T}) - \text{KL}\big(q(\mathbf{x}_T \mid \mathbf{x}_0)\|p_\theta(\mathbf{x}_T)\big) - \mathcal{L}_T \tag{7}$$

where $\mathcal{L}_T = \sum_{t=2}^{T} \mathbb{E}_{q(\mathbf{x}_{t:T}|\mathbf{x}_0)} \text{KL}(q(\mathbf{x}_{t-1} \mid \mathbf{x}_0)\|p_\theta(\mathbf{x}_{t-1} \mid \mathbf{x}_{t:T}))$. The second term is constant for most forward kernels. For specific kernel choices (e.g. absorbing or uniform), the ELBO simplifies further:

**Proposition 3.2.** *Suppose the non-Markovian diffusion process adopt an absorbing marginal kernel* $q(\boldsymbol{x}_t \mid \boldsymbol{x}_0) = \text{Cat}\big(\mathbf{x}_t; \mathbf{x}_0 \bar{\mathbf{Q}}_t\big)$, *where* $\bar{\mathbf{Q}}_t = (1 - \alpha_t)\,\mathbf{I} + \alpha_t \mathbf{1} e_m^\top$ *and* $\alpha_t$ *is a increasing function with* $\alpha_0 \approx 0$ *and* $\alpha_T \approx 1$. *The ELBO loss in Equation* (7) *can be further simplified to (see Appendix A for derivation):*

$$\mathcal{L}_{absorb} = \mathbb{E}_{\mathbf{x}_{1:T} \sim q\big(\mathbf{x}_{1:T}|\mathbf{x}_0\big)} \sum_{t=1}^{T} \big[\alpha_{t-1}\mathbf{x}_0^\top \log \mu_\theta(\mathbf{x}_{t:T}, t)\big]. \tag{8}$$

The above proposition shows that for absorbing-type forward kernels. training reduces to a weighted cross-entropy between the model's prediction and the ground-truth clean input $\mathbf{x}_0$, with time-dependent weights $\alpha_{t-1}$ reflecting the degree of corruption.

# 4 CaDDi: Causal Discrete Diffusion Model

In natural language and image modeling, decoder-only causal models have demonstrated strong performance in autoregressive modeling, due to their efficient parallel training and scalability [42, 11, 9, 43]. Meanwhile, the reverse process in non-Markovian diffusion models is inherently autoregressive: we decode the entire sequence of latent states $\mathbf{x}_{t:T}$ to retrieve $\mathbf{x}_{t-1}$. Building on this observation, we introduce **CaDDi**, a causal discrete diffusion model that unifies the *sequential* dimension (i.e., token order) and the *temporal* dimension (i.e., discrete diffusion timesteps) within a single Transformer. Specifically, CaDDi employs a standard left-to-right architecture while conditioning on multiple timesteps from the diffusion chain, enabling it to model non-Markovian discrete diffusion in a unified framework.

## 4.1 Unified Sequential and Temporal Modeling

To accommodate both sequential and temporal dependencies in a single model, a straightforward choice is to construct a data instance as a non-Markovian forward trajectory:

$$\big(\mathbf{x}_T^{(0)}, \dots, \mathbf{x}_T^{(L)}, \ \mathbf{x}_{T-1}^{(0)}, \dots, \mathbf{x}_0^{(L)}\big),$$

where the upper index $(i)$ denotes the token position in the original sequence of length $L$, and the subscript denotes the diffusion timestep. As illutrated in Fig. 2, block-wise causal mask is then applied, where we can use the logits from the last block (i.e. block corresponding to the position of $\mathbf{x}_t$) as the prediction $\mathbf{x}_0$ of the denoiser, assuming a $\mathbf{x}_0$-parameterization. In inference, we do the *block-wise autoregression* where predicted clean data $\widetilde{\mathbf{x}}_0$ is used for constructing the next latent variable $\mathbf{x}_{t-1}$.

**2D Rotary Positional Encoding.** While it's straightforward to use a decoder-only causal model for modeling the sequential and temporal dependency of latent chains. One of the issue is that modern causal language model typically encodes *only* the sequential dimension, via rotary positional encodings [53]. However, for non-Markovian discrete diffusion, we must capture not just the standard token-level sequence but also a temporal dimension corresponding to diffusion timesteps. To address this, we extend the original 1D rotary scheme to a 2D variant.

Specifically, standard RoPE in modern language models [3] rotates a subset of the query/key dimensions according to the token position $i$. If $\mathbf{R}^{(i)}$ denotes the rotation matrix parameterized by $i$, the attention weight between positions $i$ and $j$ becomes $\left(\mathbf{R}^{(i)}\mathbf{q}^{(i)}\right)^{\top}\left(\mathbf{R}^{(j)}\mathbf{k}^{(j)}\right)$, where $\mathbf{q}^{(i)}, \mathbf{k}^{(j)}$ are the query/key vectors, and $\mathbf{R}^{(i)}$ applies 2D rotations defined by position index $i$ to corresponding pairs of dimensions. To incorporate the timestep $t$, we introduce an additional rotation along a disjoint subspace of the embedding. This results in a block-diagonal like rotation matrix:

$$\mathbf{R}_t^{(i)} = \left[ \begin{array}{cc} \mathbf{R}_{\text{seq}}^{(i)} & 0 \\ 0 & \mathbf{R}_{\text{time}}^{(t)} \end{array} \right] \tag{9}$$

where $\mathbf{R}_{\text{seq}}^{(i)}$ is the valid part of position-based rotation matrix as before, and $\mathbf{R}_{\text{time}}^{(t)}$ applies the same rotational principle to a separate set of dimensions using the timestep $t$. By interleaving temporal based rotation in these additional dimensions, it's easy to observe that when two tokens share the same timepoints $t$:

$$\left(\mathbf{R}_t^{(i)}\mathbf{q}_t^{(i)}\right)^{\top}\left(\mathbf{R}_t^{(j)}\mathbf{k}_t^{(j)}\right) = \mathbf{q}_t^{(i)^{\top}}\mathbf{R}_0^{(j-i)}\mathbf{k}_t^{(j)} = \left(\mathbf{R}^{(i)}\mathbf{q}_t^{(i)}\right)^{\top}\left(\mathbf{R}^{(j)}\mathbf{k}_t^{(j)}\right)$$

which means that in the same timepoint the sequential attention pattern is identical to that of a conventional causal model, and $\mathbf{R}_t^{(i)}$ reduces to the usual (1D) rotation in $i$.

In practice, feeding all timesteps into the model can be prohibitively large. Thus we have proposed several optional strategies such as latent truncation and trajectory re-composition to compress latent $\mathbf{x}_{t:T}$ into fixed context window size. Details can be found in Appendix C

## 4.2 CaDDi-AR: Factorization over Token Space

As dicussed in [23, 63, 62], independent factorization over dimensions in one reverse step can make $p_\theta\left(\mathbf{x}_{t-1} \mid \mathbf{x}_{t:T}\right)$ neglecting the token dependence and fail to match the true posterior $q\left(\mathbf{x}_{t-1} \mid \mathbf{x}_{t:T}\right)$. A straightforward solution following [23] is to further factorizing over token space - leading to a token-level autoregression $p_\theta\left(\mathbf{x}_{t-1} \mid \mathbf{x}_{t:T}\right) = \prod_{i=0}^{L} p_\theta\left(\mathbf{x}_{t-1}^i \mid \mathbf{x}_{t-1}^{0:i-1}, \mathbf{x}_{t:T}\right)$ and next-token prediction loss. We denote this varient as CaDDi-AR. Notice it naturally fits our decoder-only causal language model, where the sampling process essentially correpond to autoregressive generation with historical trajectory $\mathbf{x}_{t:T}$ serving as "prompt". The token-level autoregression decomposing enables CaDDi-AR a more fine-grained granularity in per-step generation.

**Semi-Speculative Decoding** CaDDi-AR usually gives stronger performance compared with vanilla CaDDi. However, naive generation can be much slower as it requires $\mathcal{O}(L \times T)$ function evaluations for a sequence of length $L$ over $T$ timesteps. By leveraging the unique properties of decoder-only causal language modeling, we propose a *semi-speculative decoding* strategy that substantially reduces inference time while maintaining generation quality.

Specifically, although causal language models generate tokens sequentially, they can verify the probabilities of any *pre-drafted* sequence in parallel. Since CaDDi-AR shares the same denoising target $\mathbf{x}_0$ across all timesteps, this suggests a natural procedure: reuse the previous timestep's predictions $\widetilde{\mathbf{x}}_0^{\text{prev}}$ as a *draft* for the current timestep (see Algorithm 2). The model then *verifies* these drafted tokens in parallel, accepting those that meet a specified confidence threshold (e.g., high probability).

---

**Algorithm 2** Semi-Speculative Decoding for CaDDi-AR

1: **Input:** model parameters $\theta$, prior distribution $q(\mathbf{x}_T)$
2: **Output:** Sampled data $\mathbf{x}_0$
3: **for** $t = T$ **down to** 1 **do**
4:    $i \leftarrow 0$
5:    **if** $\widetilde{\mathbf{x}}_0^{\text{prev}}$ is available **then**
6:       $i \leftarrow \text{VERIFY}(p_\theta, \mathbf{x}_{t:T}, \widetilde{\mathbf{x}}_0^{\text{prev}})$
7:       $\widetilde{\mathbf{x}}_0^{i,\text{prev}} \leftarrow \text{CORRECT}(p_\theta, \mathbf{x}_{t:T}, \widetilde{\mathbf{x}}_0^{\text{prev}})$
8:       $\widetilde{\mathbf{x}}_0^{0:i} \leftarrow \widetilde{\mathbf{x}}_0^{0:i,\text{prev}}$
9:    **while** $i < L$ **do**
10:       $\widetilde{\mathbf{x}}_0^{i+1} \leftarrow p_\theta\left(\mathbf{x}_0 \mid \mathbf{x}_{t:T}, \widetilde{\mathbf{x}}_0^{0:i}\right)$
11:       $i \leftarrow i + 1$
12:    $\mathbf{x}_{t-1} \sim q\left(\mathbf{x}_{t-1} \mid \widetilde{\mathbf{x}}_0\right)$
13:    $\widetilde{\mathbf{x}}_0^{\text{prev}} \leftarrow \widetilde{\mathbf{x}}_0$
14: **return** $\mathbf{x}_0$

---

Figure 3: Semi-Speculative Decoding with CaDDi-AR: The model verifies all tokens in parallel to identify the first rejection index $i$, then resumes sampling from that point.

This approach closely resembles speculative decoding [34, 8], with one key difference: we do not rely on a separate, smaller model to propose the draft sequence. Instead, CaDDi-AR's own predictions

from the preceding timestep serve as the draft. Like speculative decoding, various verification and correction strategies (e.g., greedy, nucleus sampling) can be employed, ensuring either a comparable or identical sampling distribution while significantly reducing the total number of sampling steps.

### 4.3 Leveraging Pretrained LLM for Discrete Diffusion

One key observation is that standard causal language modeling can be seen as a **special case** of CaDDi-AR under particular settings: specifically, when using a single-step diffusion process ($T = 1$) without any latent trajectory conditioning as 'prompt', i.e., $p_\theta \left( \mathbf{x}_0^i \mid \mathbf{x}_0^{0:i-1}, \varnothing \right)$. Furthermore, as shown in Section 4.1, the 2D rotary positional encoding can be seamlessly integrated into existing language models, enabling a unified treatment of both sequential and temporal dimensions..

Given these equivalences, one can take a pretrained LLM (trained in a standard causal manner) and further fine-tune it with the CaDDi-AR diffusion objective[4]. By allowing the model to condition on historical latent variables $\mathbf{x}_{t+1:T}$, we endow it with iterative denoising and bidirectional modeling capabilities. This straightforward adaptation expands the model's generation modes (e.g., text infilling) while fully leveraging the pretrained language model's existing knowledge.

While several recent works have fine-tuned autoregressive LLMs as discrete diffusion models [64, 54, 6, 22], our approach preserves the natural causal masking across both temporal and token dimensions. This design choice significantly mitigates the model drift issues observed in prior methods that replaced causal masks with full attention, which often required careful learning rate tuning to stabilize training [64].

## 5 Experiments

**Baseline.** For most of our benchmark we compare CaDDi against several well-established discrete diffusion models: D3PM [2], SEDD [37], MDLM [44], UDLM [46], and Discrete Flow Matching [21]. We adopt the official MDLM and UDLM codebases, which also include reference implementations of D3PM and SEDD. All models use a 12-layer Transformer (hidden size 768, 12 attention heads), trained with a learning rate of 3e-4, 2500 warm-up steps, and linear learning rate annealing. For evaluation, we sample the same number of sequences across all models. On the Text8 benchmark, we additionally include flow-based methods IAF/SCF [67], Argmax Flow [31], Discrete Flow [58], Any-order Autoregressive Models [30], MAC [49], and Mult. Diffusion [31]. We denote our block-level autoregressive variant as CaDDi and the token-level variant as CaDDi-AR. Unless otherwise noted, all models are trained from scratch and experiments by default use absorbing-marginal kernels, corresponding to the linear noise schedule of D3PM [2].

Table 1: Evaluation on the LM1B dataset. We report guided Generative Perplexity (PPL) under three pretrained causal language models (GPT-2, Llama-2-7B, and Llama-3.2-3B) at different sampling temperatures of each baseline model and CaDDi ($T = 1$, $T = 0.7$, $T = 0.5$). The best performance is **bolded**, and the second-best is underlined.

| Model | GPT2 | | | LLAMA-2 | | | LLAMA-3 | | | ENTROPY |
|---|---|---|---|---|---|---|---|---|---|---|
| | T=1 | T=0.7 | T=0.5 | T=1 | T=0.7 | T=0.5 | T=1 | T=0.7 | T=0.5 | |
| Data | 114.79 | — | — | 22.97 | — | — | 42.20 | — | — | — |
| UDLM | 313.24 | 318.80 | 328.99 | 110.37 | 111.86 | 119.21 | 207.91 | 219.86 | 231.05 | 6.0 |
| D3PM | 232.37 | 134.55 | 133.38 | 76.00 | 55.54 | 59.02 | 145.96 | 98.26 | 110.86 | 5.7 |
| SEDD | 201.19 | 148.43 | 81.44 | 77.54 | 66.54 | 46.91 | 144.23 | 84.20 | 60.00 | 5.6 |
| MDLM | 199.45 | 135.85 | 106.20 | 67.86 | 62.34 | 60.09 | 126.35 | 113.19 | 104.71 | 5.6 |
| DFM | 182.21 | 94.46 | 106.03 | 67.02 | 38.93 | 61.91 | **120.09** | 66.22 | 102.89 | 5.7 |
| **CaDDi** | 142.51 | **64.27** | **45.96** | 70.27 | 35.40 | **23.81** | 124.51 | **58.26** | **36.79** | 5.7 |
| **CaDDi-AR** | **139.80** | 76.02 | 67.59 | **65.93** | **35.38** | 27.76 | 121.25 | 59.66 | 44.54 | 5.7 |

**One Billion Words Dataset** We evaluate CaDDi's generative capabilities on the One Billion Words dataset (LM1B) [7], a large-scale natural language corpus comprising over 30 million English sen-

---

[4]Pretrained causal LLM can also be fine-tuned as vanilla CaDDi models, using a simple causal bidirectional augmentation trick. See Appendix C.

tences of varying lengths. We follow the tokenization and training setup introduced in DiffusionBERT [26]. For UDLM, we use the pretrained weights provided by the authors. For other baselines, we retrain the models using their official codebases, as checkpoints are not publicly available.

To evaluate the quality of model-generated text, we report generative perplexity (Gen PPL), computed using several large language models as oracles, including GPT-2 [42], Llama-2 (7B) [57], and Llama-3 (3B) [18]. All oracle models are pretrained on large-scale natural language corpora. To assess the diversity of generated outputs, we additionally compute the entropy of the generated text set without applying temperature scaling. Further details on these metrics are provided in Appendix E.1.

As shown in Table 1, CaDDi-AR and CaDDi consistently outperform baselines in generative perplexity across the three language model oracles. Notably, at low temperature, CaDDi alone performs comparably to—or even better than—CaDDi-AR. We hypothesize that this is due to the long-tail issue being more pronounced in block generation with fewer steps, where lower temperature helps mitigate sampling noise. Importantly, our approach maintains diversity, as indicated by comparable entropy scores.

**Text8 Dataset**   Following prior work [2, 48], we trained a discrete diffusion model on short text chunks of length 256 from the text8 dataset. We use the same dataset split as in previous studies, training on the training set and reporting performance on the test set using the standard bits-per-dimension (BPD) metric. The BPD is defined as: $-\frac{1}{L}\sum_{i=1}^{L}\log_2 p(\mathbf{x}_i)$. Note that the bound of log-likelihood depends on the discretization of the diffusion schedule. To enable fair comparisons, we report the number of discretization steps used for each diffusion-based model.

We omit BPD results for CaDDi-AR due to the additional approximation error of likelihood introduced by its token-level autoregressive decomposition. As shown in Table 2, under the same discretization setting (64 steps), CaDDi outperforms other discrete diffusion models. While any-order autoregressive models can be viewed as a special case of discrete diffusion with 256 steps, our 64-step model already approaches their performance.

Table 2: **Evaluation results on Text8 Dataset** reporting layers, steps/discretization, bits-per-dimension (BPD), perplexity, and negative log-likelihood (NLL) for all compared models. For diffusion-based and any-order models, the values represent variational upper bounds. Rows shaded in gray correspond to evaluations at 64 steps, enabling a direct comparison with CaDDi.

| Model | Layers | Steps/Discretization | BPD↓ | Perplexity↓ | NLL↓ |
|---|---|---|---|---|---|
| *Autoregressive* | | | | | |
| IAF/SCF | 12 | - | 1.88 | 3.68 | 1.30 |
| Argmax Flow | 12 | - | 1.39 | 2.62 | 0.96 |
| Discrete Flow | 3×8 | - | **1.23** | **2.35** | **0.85** |
| Autoregressive | 12 | - | **1.23** | **2.35** | **0.85** |
| *Any Order Autoregressive* | | | | | |
| ARDM | 12 | - | ≤1.43 | ≤2.69 | ≤0.99 |
| MAC | 12 | - | ≤1.40 | ≤2.64 | ≤0.97 |
| *Diffusion* | | | | | |
| Mult. Diffusion | 12 | 1000 | ≤1.72 | ≤3.29 | ≤1.19 |
| D3PM Absorb | 12 | ∞ (continuous) | ≤1.45 | ≤2.73 | ≤1.01 |
| | 12 | 64 | ≤1.51 | ≤2.85 | ≤1.05 |
| SEDD Absorb | 12 | ∞ (continuous) | ≤1.39 | ≤2.62 | ≤0.96 |
| | 12 | 64 | ≤1.46 | ≤2.75 | ≤1.01 |
| UDLM | 12 | ∞ (continuous) | ≤1.44 | ≤2.71 | ≤1.00 |
| | 12 | 64 | ≤1.60 | ≤3.03 | ≤1.11 |
| MDLM | 12 | ∞ (continuous) | ≤1.40 | ≤2.64 | ≤0.97 |
| | 12 | 64 | ≤1.46 | ≤2.75 | ≤1.01 |
| CaDDi | 12 | 64 | **≤1.41** | **≤2.66** | **≤0.98** |

**General Reasoning Datasets with Fine-tuned LLM**   As shown in Section 4.3, CaDDi-AR retains compatibility with standard causal language models and can be fine-tuned from pretrained LLM checkpoints without architectural modifications. To evaluate its general reasoning ability, we fine-

Table 3: Accuracy (%) on a range of natural language reasoning benchmarks.

| Model | Size | ARC-Chal. | ARC-Easy | BoolQ | PIQA | RACE | Social IQA | LAMBADA |
|---|---|---|---|---|---|---|---|---|
| QWen2-1.5B | 1.5B | 33.7 | 66.1 | **72.6** | 75.4 | **36.6** | 45.8 | 63.9 |
| GPT-2 | 1.5B | 28.5 | 51.1 | 61.8 | 70.5 | 33.1 | 40.3 | 44.6 |
| TinyLlama | 1.1B | 29.9 | 52.2 | 59.4 | 70.3 | 35.6 | 39.4 | 43.2 |
| MDM | 1.1B | - | 48.7 | 62.2 | 69.5 | 35.6 | 41.0 | 52.7 |
| CaDDi-AR | 1.5B | **34.2** | **67.8** | 71.6 | 75.4 | 34.3 | **46.9** | **66.3** |

tune a 1.5B QWen model using the CaDDi-AR diffusion objective on a range of natural language understanding benchmarks, including ARC-Challenge, ARC-Easy [14], BoolQ [13], PIQA [4], RACE [33], Social IQA [45], and LAMBADA [40]. These tasks span multiple reasoning paradigms, from commonsense inference to multi-hop reading comprehension. We benchmark with model of comparable size, including causal language model QWen2 [1], GPT2 [42], TinyLlama [65] and a discrete diffusion model MDM [39] trained from scratch.

As shown in Table 3, CaDDi-AR achieves consistent gains over diffusion-based baselines of comparable size. It also include +1.9% improvement on ARC-Challenge and a +2.4% gain on LAMBADA over the base QWen model. These results suggest that CaDDi-AR's non-Markovian formulation supports more robust and revisable reasoning, while maintaining compatibility with existing language model infrastructures. It's worth noting that compared with MDM, CaDDi-AR requires much less training effort.

**Conditional Text Generation** We also evaluate conditional text generation on the Amazon Polarity dataset [38], which consists of 3.6M Amazon reviews labeled as positive or negative. We adapt this task as text infilling by prepending a label-based prompt to each review (see Appendix E.2 for details). We train the conditional generator $p_\theta(\mathbf{x}_0 \mid \mathbf{x}_t, y)$ alongside unconditional one $p_\theta(\mathbf{x}_0 \mid \mathbf{x}_t)$, by preserving certain parts of the text as fixed.

Table 4: Comparison of GPT-2, CaDDi, and CaDDi-CFG on the Amazon Polarity dataset. Generation is conditioned on either positive or negative.

| Model | Condition | Sentiment Accuracy (%) |
|---|---|---|
| **GPT-2** | Positive | 73.07 |
| | Negative | 75.18 |
| **CaDDi-CFG**$_{\gamma=1}$ | Positive | 71.37 |
| | Negative | 85.42 |
| **CaDDi-CFG**$_{\gamma=1.25}$ | Positive | 73.61 |
| | Negative | 85.92 |

We measure sentiment accuracy (SA) using a fine-tuned DistilBERT classifier. As shown in Table 4, our approach achieves performance comparable to a fine-tuned GPT-2 on the same dataset while offering more flexible generation (unlike GPT2 only allow prompting from begining, our method allows for prompting from arbitrary parts of the text, such as the middle or the title. Examples are shown in Figure 11). Furthermore, by applying classifier-free guidance (denoted as CaDDi-CFG) [27, 46] with different guidance scales $\gamma$, we generate reviews that better align with the given prompts

**Additional Analysis and Ablation Study.** To assess the inference robustness of discrete diffusion models, we perform an additional ablation study by manually injecting controlled noise (i.e., incorrect predictions) at various timesteps during generation, simulating potential inference-time errors. As shown in Figure 4, introducing noise at earlier stages generally leads to greater error accumulation. Nonetheless, our proposed CaDDi framework consistently exhibits stronger resilience compared to D3PM and MDLM, maintaining higher generation quality under perturbations across all settings. We remain the full ablation study and discussion in Appendix G.

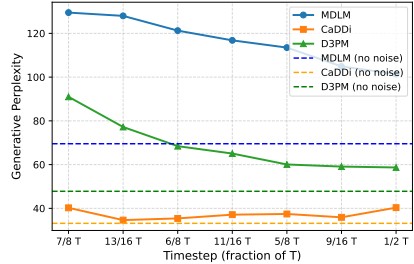

Figure 4: Generation performance under manually injected noise at different timestep

## 6 Conclusion

We introduced *CaDDi*, a causal discrete diffusion framework that relaxes the traditional Markovian assumptions in favor of an autoregressive inference process. By explicitly conditioning each denoising step on the entire future trajectory, CaDDi captures richer temporal dependencies and leverages

iterative refinement. Critically, our approach can also be built atop existing causal language models—bridging standard sequence modeling with powerful diffusion capabilities—while preserving both knowledge from large-scale training and the flexibility of iterative editing.

## Acknowledgement

The authors thank collaborators and contributors from across institutions for their invaluable support and insights throughout this project. This work was supported in part by the National Institutes of Health (NIH) grant R35GM143072–01, the National Science Foundation (NSF) grant IIS Division of Information & Intelligent Systems 2443528, and the Yale Colton Center Award, all awarded to Prof. David van Dijk; and by the National Science Foundation (NSF) grants IIS Division of Information & Intelligent Systems 2403317 and CNS Division of Computer and Network Systems 2431504, as well as the Envisioning Artificial Intelligence at Yale 2025 initiative from the Yale Office of the Provost, awarded to Prof. Rex Ying.

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

# A  Proof

Throughout this appendix we work with a *single–token* random variable that takes values in the augmented vocabulary $\mathcal{V}_+ := \mathcal{V} \cup \{\texttt{[MASK]}\}$. The distinguished absorbing symbol is denoted by $m \in \mathcal{V}_+$ and we write $e_m$ for the corresponding standard basis vector.

Let $\mathbf{x}_0 \sim p_0$ be drawn from the empirical data distribution, which assigns zero probability to $m$. Fix a diffusion horizon $T \in \mathbb{N}$.

## A.1  Proof of Proposition 3.1

*An absorbing–state non–Markovian discrete diffusion process with marginal transition kernel $\bar{\mathbf{Q}}_t = (1 - \alpha_t)\mathbf{I} + \alpha_t \mathbf{1} e_m^\top$ admits a bijection to an absorbing–state Markovian discrete diffusion process with marginal transition kernel $\bar{\mathbf{Q}}_t^* = (1 - \alpha_t^*)\mathbf{I} + \alpha_t^* \mathbf{1} e_m^\top$ such that the two processes exhibit identical mutual–information decay between $\mathbf{x}_{t:T}$ and $\mathbf{x}_0$, provided that the coefficients satisfy $\alpha_t^* = \prod_{\tau=t}^{T} \alpha_\tau$.*

We begin by formalising the two forward corruption processes that will be compared.

### A.1.1  Forward–process definitions

**Definition A.1** (Markovian absorbing diffusion). Let $\boldsymbol{\beta} = (\beta_1, \ldots, \beta_T) \subset [0,1]^T$ be the *single–step masking probabilities*. The Markov chain $(\mathbf{x}_s)_{s=0}^T$ is specified by

$$\mathbf{x}_s \mid \mathbf{x}_{s-1} \sim (1 - \beta_s)\delta_{\mathbf{x}_{s-1}} + \beta_s \delta_m, \qquad s = 1, \ldots, T. \tag{10}$$

Because $m$ is absorbing, the cumulative masking probability after $t$ steps is

$$\alpha_t^* := \Pr\big(\mathbf{x}_t = m \mid \mathbf{x}_0\big) = 1 - \prod_{s=1}^{t}(1 - \beta_s), \qquad t = 1, \ldots, T, \tag{11}$$

which yields the marginal transition kernel

$$\bar{\mathbf{Q}}_t^{\mathrm{M}} = (1 - \alpha_t^*)\mathbf{I} + \alpha_t^* \mathbf{1} e_m^\top.$$

**Definition A.2** (Non–Markovian absorbing diffusion). For a *noise schedule* $\boldsymbol{\alpha} = (\alpha_1, \ldots, \alpha_T) \subset [0,1]^T$ we generate the forward trajectory by

$$\mathbf{x}_s \mid \mathbf{x}_0 \overset{\text{i.i.d.}}{\sim} (1 - \alpha_s)\delta_{\mathbf{x}_0} + \alpha_s \delta_m, \qquad s = 1, \ldots, T. \tag{12}$$

Each $\mathbf{x}_s$ is obtained by independently masking $\mathbf{x}_0$ with probability $\alpha_s$. The associated marginal transition kernel is
$$\bar{\mathbf{Q}}_t^{\mathrm{NM}} = (1 - \alpha_t)\mathbf{I} + \alpha_t \mathbf{1} e_m^\top, \qquad t = 1, \ldots, T.$$

### A.1.2  Mutual–information computations and decay measure

For any pair of random variables $(U, V)$ we recall that $I(U; V) = H(U) - H(U \mid V)$. The entropy $H(\mathbf{x}_0) = -\sum_{v \in \mathcal{V}} p_0(v) \log p_0(v)$ is finite and strictly positive.

To make comparisons between Markovian and non-Markovian diffusion processes more interpretable, we introduce the **normalized mutual information decay** following [2]:

$$D_t := 1 - \frac{I(\mathbf{x}_t; \mathbf{x}_0)}{H(\mathbf{x}_0)} = \frac{H(\mathbf{x}_0, \mathbf{x}_t) - H(\mathbf{x}_t)}{H(\mathbf{x}_0)} \tag{13}$$

which quantifies the proportion of information about $\mathbf{x}_0$ that is lost after corruption step $t$.

**Lemma A.3** (Markovian suffix information). *For the process of Definition A.1 and every $t \in \{1, \ldots, T\}$,*

$$I_{\mathrm{M}}(\mathbf{x}_{t:T}; \mathbf{x}_0) = H(\mathbf{x}_0)(1 - \alpha_t^*), \quad D_t^{\mathrm{M}} = \alpha_t^* \tag{14}$$

Here we give two proofs:

*Proof 1 (Direct Expansion).* Because the chain is absorbing, the suffix $\mathbf{x}_{t:T}$ is a deterministic function of the single variable $\mathbf{x}_t$; hence

$$I_{\mathrm{M}}(\mathbf{x}_{t:T}; \mathbf{x}_0) = I(\mathbf{x}_t; \mathbf{x}_0).$$

There are two mutually exclusive outcomes:

(i) Un-masked: with probability $1 - \alpha_t^*$ the token is uncorrupted and $\mathbf{x}_t = \mathbf{x}_0$.

(ii) Masked: with probability $\alpha_t^*$ the token is replaced by the absorbing symbol $e_m$, which is independent of $\mathbf{x}_0$.

Writing $p_0(v) = \Pr(\mathbf{x}_0 = v)$, the joint pmf is

$$p(\mathbf{x}_0 = v, \mathbf{x}_t = u) = \begin{cases} (1 - \alpha_t^*)\, p_0(v), & u = v \neq e_m, \\ \alpha_t^*\, p_0(v), & u = e_m, \\ 0, & \text{otherwise.} \end{cases}$$

The corresponding marginal of $\mathbf{x}_t$ is

$$p(\mathbf{x}_t = u) = \begin{cases} (1 - \alpha_t^*)\, p_0(u), & u \neq e_m, \\ \alpha_t^*, & u = e_m. \end{cases}$$

Direct expansion of mutual information.

$$I(\mathbf{x}_t; \mathbf{x}_0) = \sum_{u,v} p(u, v) \log \frac{p(u, v)}{p(u)\, p(v)}.$$

Only two cases have non-zero probability:

(i) *Case $u = v \neq e_m$.* Here $p(u, v) = (1 - \alpha_t^*)p_0(v)$ and $p(u) = (1 - \alpha_t^*)p_0(v)$, so

$$\log \frac{p(u, v)}{p(u)p(v)} = \log \frac{(1 - \alpha_t^*)p_0(v)}{(1 - \alpha_t^*)p_0(v)\, p_0(v)} = -\log p_0(v).$$

The total contribution is

$$\sum_{v \neq e_m} (1 - \alpha_t^*)p_0(v)\big[-\log p_0(v)\big] = (1 - \alpha_t^*)H(\mathbf{x}_0).$$

(ii) *Case $u = e_m$.* In this case $p(u, v) = \alpha_t^* p_0(v)$ and $p(u) = \alpha_t^*$; the logarithmic term vanishes, so the contribution is zero.

Hence

$$I(\mathbf{x}_t; \mathbf{x}_0) = (1 - \alpha_t^*)H(\mathbf{x}_0).$$

$$D_t^{\mathrm{M}} = 1 - \frac{I(\mathbf{x}_t; \mathbf{x}_0)}{H(\mathbf{x}_0)} = 1 - (1 - \alpha_t^*) = \alpha_t^*.$$

Since $I(\mathbf{x}_{t:T}; \mathbf{x}_0) = I(\mathbf{x}_t; \mathbf{x}_0)$, the stated identities follow. $\qquad\square$

*Proof 2 (Conditional Entropy).* Condition on the observed token $\mathbf{x}_t$:

(i) If $\mathbf{x}_t = e_m$ (probability $\alpha_t^*$), masking reveals nothing, so $H(\mathbf{x}_0 \mid \mathbf{x}_t = e_m) = H(\mathbf{x}_0)$.

(ii) If $\mathbf{x}_t = v \neq e_m$ (probability $1 - \alpha_t^*$), we know $\mathbf{x}_0 = v$ exactly, hence $H(\mathbf{x}_0 \mid \mathbf{x}_t = v) = 0$.

Averaging,
$$H(\mathbf{x}_0 \mid \mathbf{x}_t) = \alpha_t^* H(\mathbf{x}_0) + (1 - \alpha_t^*) \cdot 0 = \alpha_t^* H(\mathbf{x}_0).$$

Therefore
$$I(\mathbf{x}_t; \mathbf{x}_0) = H(\mathbf{x}_0) - H(\mathbf{x}_0 \mid \mathbf{x}_t) = (1 - \alpha_t^*) H(\mathbf{x}_0), \qquad D_t^{\mathrm{M}} = 1 - \frac{I(\mathbf{x}_t; \mathbf{x}_0)}{H(\mathbf{x}_0)} = \alpha_t^*.$$

Because $\mathbf{x}_{t:T}$ is a deterministic function of $\mathbf{x}_t$, the same formula holds for $I(\mathbf{x}_{t:T}; \mathbf{x}_0)$. $\qquad\square$

**Lemma A.4** (Non–Markovian suffix information)**.** *For the process of Definition A.2 and every* $t \in \{1, \ldots, T\}$,

$$I_{\mathrm{NM}}(\mathbf{x}_{t:T}; \mathbf{x}_0) = H(\mathbf{x}_0)\left(1 - \prod_{\tau=t}^{T} \alpha_\tau\right), \qquad D_t^{\mathrm{NM}} = \prod_{\tau=t}^{T} \alpha_\tau \qquad (15)$$

*Proof.* Define the indicator $Z := \mathbf{1}\{\mathbf{x}_\tau = e_m \text{ for every } \tau = t, \ldots, T\}$. Conditioned on $\mathbf{x}_0$, each coordinate is masked independently with probability $\alpha_\tau$, so

$$\Pr(Z = 1) = \prod_{\tau=t}^{T} \alpha_\tau, \qquad \Pr(Z = 0) = 1 - \prod_{\tau=t}^{T} \alpha_\tau.$$

The event $Z$ depends only on the masking coins, hence is independent of the value of $\mathbf{x}_0$.

For conditional entropy $H(\mathbf{x}_0 \mid \mathbf{x}_{t:T})$.

(i) If $Z = 1$ (all tokens masked, probability $\prod \alpha_\tau$), the suffix reveals nothing, so $H(\mathbf{x}_0 \mid \mathbf{x}_{t:T}) = H(\mathbf{x}_0)$.

(ii) If $Z = 0$ (at least one un-masked coordinate, probability $1 - \prod \alpha_\tau$), every un-masked coordinate equals $\mathbf{x}_0$; thus $\mathbf{x}_0$ is known exactly and $H(\mathbf{x}_0 \mid \mathbf{x}_{t:T}) = 0$.

Averaging over $Z$,

$$H(\mathbf{x}_0 \mid \mathbf{x}_{t:T}) = \left(\prod_{\tau=t}^{T} \alpha_\tau\right) H(\mathbf{x}_0).$$

Using $I(U; V) = H(U) - H(U \mid V)$,

$$I_{\mathrm{NM}}(\mathbf{x}_{t:T}; \mathbf{x}_0) = H(\mathbf{x}_0) - H(\mathbf{x}_0 \mid \mathbf{x}_{t:T}) = H(\mathbf{x}_0)\left(1 - \prod_{\tau=t}^{T} \alpha_\tau\right).$$

$$D_t^{\mathrm{NM}} = 1 - \frac{I_{\mathrm{NM}}(\mathbf{x}_{t:T}; \mathbf{x}_0)}{H(\mathbf{x}_0)} = \prod_{\tau=t}^{T} \alpha_\tau.$$

$\qquad\square$

These results show that both the Markovian and non-Markovian processes exhibit the same normalized mutual information decay if their cumulative masking curves satisfy:

$$\alpha_t^* = \prod_{\tau=t}^{T} \alpha_\tau. \qquad (16)$$

Since $\alpha_t^*$ and $\beta_t$ have correspondence as shown in Equation 11, one can easily further derive:

**Proposition A.5** (Equivalence in mutual-information decay). *Let $\boldsymbol{\alpha} \subset [0,1]^T$ be a non–Markovian schedule and define the effective cumulative schedule $\alpha_t^* := \prod_{\tau=t}^T \alpha_\tau$. Then there exists a Markovian schedule $\boldsymbol{\beta} \subset [0,1]^T$ defined by*

$$\beta_t := 1 - \frac{1 - \alpha_t^*}{1 - \alpha_{t-1}^*}, \qquad \alpha_0^* := 0, \tag{17}$$

*such that:*

$$D_t^{\mathrm{NM}} = D_t^{\mathrm{M}} \quad and \quad I_{\mathrm{NM}}\left(\mathbf{x}_{t:T}; \mathbf{x}_0\right) = I_{\mathrm{M}}\left(\mathbf{x}_{t:T}; \mathbf{x}_0\right), \tag{18}$$

*for all $t = 1, \dots, T$. This ensures equivalence in both absolute and relative information loss.*

**Matching the linear marginal used in prior work.** Most existing studies on absorbing-state Markovian diffusions [2, 37, 44] adopt the *linear* cumulative–masking curve $\alpha_t^{\mathrm{lin},*} = t/T$, $t = 0, \dots, T$. For a fair comparison by default we employ a non-Markovian *independent* schedule $\alpha^{\mathrm{lin}}$ that reproduces exactly the same marginals in our experiments, i.e. satisfies $\alpha_t^{\mathrm{lin},*} = \prod_{\tau=t}^T \alpha_\tau^{\mathrm{lin}}$. Using the backward recursion $\alpha_t = \alpha_t^* / \alpha_{t+1}^*$ (Sec. A.1) gives the closed-form

$$\alpha_t^{\mathrm{lin}} = \frac{t}{t+1}, \quad t = 1, \dots, T-1, \qquad \alpha_T^{\mathrm{lin}} = 1$$

## A.2 Derivation of the Non-Markovian Evidence Lower Bound (ELBO)

We derive a variational lower bound on the marginal log-likelihood of the observed data $\mathbf{x}_0$ under the non-Markovian discrete diffusion model.

We start with the marginal likelihood of the observed data:

$$\log p_\theta\left(\mathbf{x}_0\right) = \log \int p_\theta\left(\mathbf{x}_0, \mathbf{x}_{1:T}\right) d\mathbf{x}_{1:T} = \log \int \frac{p_\theta\left(\mathbf{x}_0, \mathbf{x}_{1:T}\right)}{q\left(\mathbf{x}_{1:T} \mid \mathbf{x}_0\right)} q\left(\mathbf{x}_{1:T} \mid \mathbf{x}_0\right) d\mathbf{x}_{1:T}$$

Applying Jensen's inequality yields the Evidence Lower Bound (ELBO):

$$\log p_\theta\left(\mathbf{x}_0\right) \geq \mathbb{E}_{q\left(\mathbf{x}_{1:T} \mid \mathbf{x}_0\right)} \left[\log \frac{p_\theta\left(\mathbf{x}_0, \mathbf{x}_{1:T}\right)}{q\left(\mathbf{x}_{1:T} \mid \mathbf{x}_0\right)}\right] = \mathcal{L}_{\text{non-markov}}$$

In the non-Markovian case, the joint distribution over the reverse generative process is:

$$p_\theta\left(\mathbf{x}_0, \mathbf{x}_{1:T}\right) = p_\theta\left(\mathbf{x}_T\right) \prod_{t=1}^T p_\theta\left(\mathbf{x}_{t-1} \mid \mathbf{x}_{t:T}\right)$$

where $\mathbf{x}_0$ is generated at the final step. The approximate posterior, due to the independent corruption assumption, factorizes as:

$$q\left(\mathbf{x}_{1:T} \mid \mathbf{x}_0\right) = \prod_{t=1}^T q\left(\mathbf{x}_t \mid \mathbf{x}_0\right)$$

Finally, we have the ELBO expansion:

$$
\begin{aligned}
\mathcal{L}_{\text{non-markov}} &= \mathbb{E}_{q\left(\mathbf{x}_{1:T} \mid \mathbf{x}_0\right)} \left[\log \frac{p_\theta(\mathbf{x}_T)}{q(\mathbf{x}_T \mid \mathbf{x}_0)} + \sum_{t=2}^T \log \frac{p_\theta(\mathbf{x}_{t-1} \mid \mathbf{x}_{t:T})}{q(\mathbf{x}_{t-1} \mid \mathbf{x}_0)} + \log p_\theta(\mathbf{x}_0 \mid \mathbf{x}_{1:T})\right] \\
&= \underbrace{\mathbb{E}_{q\left(\mathbf{x}_{1:T} \mid \mathbf{x}_0\right)} \log p_\theta(\mathbf{x}_0 \mid \mathbf{x}_{1:T})}_{\text{Reconstruction}} - \underbrace{\mathrm{KL}\left(q(\mathbf{x}_T \mid \mathbf{x}_0) \parallel p_\theta(\mathbf{x}_T)\right)}_{\text{Prior KL}} \\
&\quad - \underbrace{\sum_{t=2}^T \mathbb{E}_{q\left(\mathbf{x}_{t:T} \mid \mathbf{x}_0\right)} \mathrm{KL}\left(q(\mathbf{x}_{t-1} \mid \mathbf{x}_0) \parallel p_\theta(\mathbf{x}_{t-1} \mid \mathbf{x}_{t:T})\right)}_{\text{Reverse KLs}}
\end{aligned}
\tag{19}
$$

We can denote the accumulated reverse KL terms as:

$$\mathcal{L}_T = \sum_{t=2}^{T} \mathbb{E}_{q(\mathbf{x}_{t:T}|\mathbf{x}_0)} \, \mathrm{KL} \left( q \left( \mathbf{x}_{t-1} \mid \mathbf{x}_0 \right) \| p_\theta \left( \mathbf{x}_{t-1} \mid \mathbf{x}_{t:T} \right) \right) \tag{20}$$

With this notation, the non-Markovian ELBO simplifies to Equation 7:

$$\mathcal{L}_{\text{non-markov}} = \mathbb{E}_{q(\mathbf{x}_{1:T}|\mathbf{x}_0)} \log p_\theta \left( \mathbf{x}_0 \mid \mathbf{x}_{1:T} \right) - \mathrm{KL} \left( q \left( \mathbf{x}_T \mid \mathbf{x}_0 \right) \| p_\theta \left( \mathbf{x}_T \right) \right) - \mathcal{L}_T \tag{21}$$

Note that the second term, corresponding to the prior KL, is constant for most diffusion kernels and can be omitted during training.

### A.3 Proof of Proposition 3.2

*Suppose the non-Markovian diffusion process adopt an absorbing marginal kernel $q \left( \mathbf{x}_t \mid \mathbf{x}_0 \right) = \mathrm{Cat} \left( \mathbf{x}_t; \mathbf{x}_0 \bar{\mathbf{Q}}_t \right)$, where $\bar{\mathbf{Q}}_t = (1 - \alpha_t) \, \mathbf{I} + \alpha_t \mathbf{1} e_m^\top$ and $\alpha_t$ is a increasing function with $\alpha_0 \approx 0$ and $\alpha_T \approx 1$. The ELBO loss in Equation (7) can be further simplified to:*

$$\mathcal{L}_{absorb} = \mathbb{E}_{\mathbf{x}_{1:T} \sim q \left( \mathbf{x}_{1:T}|\mathbf{x}_0 \right)} \sum_{t=1}^{T} \left[ \alpha_{t-1} \mathbf{x}_0^\top \log \mu_\theta (\mathbf{x}_{t:T}, t) \right].$$

**Proof.** Suppose the non-Markovian diffusion process adopts an absorbing marginal kernel $\overline{\mathbf{Q}}_t = (1 - \alpha_t) \, \mathbf{I} + \alpha_t \mathbf{1} e_m^\top$, we have

$$q \left( \mathbf{x}_t | \mathbf{x}_0 \right) = \begin{cases} \alpha_t & \mathbf{x}_t = e_m \\ 1 - \alpha_t & \mathbf{x}_t = \mathbf{x}_0 \\ 0 & \text{otherwise.} \end{cases} \tag{22}$$

We now evaluate the per-step KL term in the ELBO 19:

$$\begin{aligned} \mathrm{KL} \left( q(\mathbf{x}_{t-1} \mid \mathbf{x}_0) \| p_\theta(\mathbf{x}_{t-1} \mid \mathbf{x}_{t:T}) \right) &= \mathrm{KL} \left( q(\mathbf{x}_{t-1} \mid \mathbf{x}_0) \| q(\mathbf{x}_{t-1} \mid \mathbf{x}_{t+1}, \mu_\theta(\mathbf{x}_{t:T}, t)) \right) \\ &= \underbrace{q(\mathbf{x}_{t-1} = e_m \mid \mathbf{x}_0) \log \frac{q(\mathbf{x}_{t-1} = e_m \mid \mathbf{x}_0)}{q(\mathbf{x}_{t-1} = e_m \mid \mu_\theta(\mathbf{x}_{t:T}, t))}}_{0} \\ &\quad + \sum_{k \neq m} q(\mathbf{x}_{t-1} = e_k \mid \mathbf{x}_0) \log \frac{q(\mathbf{x}_{t-1} = e_k \mid \mathbf{x}_0)}{q(\mathbf{x}_{t-1} = e_k \mid \mu_\theta(\mathbf{x}_{t:T}, t))} \\ &= \sum_{k \neq m} \alpha_{t-1} \mathbf{x}_0^\top e_k \log \frac{\mathbf{x}_0^\top e_k}{\left[ \mu_\theta(\mathbf{x}_{t:T}, t) \right]^\top e_k} \\ &= -\alpha_{t-1} \mathbf{x}_0^\top \log \mu_\theta(\mathbf{x}_{t:T}, t) \end{aligned} \tag{23}$$

Next, consider the reconstruction term:

$$\begin{aligned} \mathbb{E}_{q(\mathbf{x}_{1:T}|\mathbf{x}_0)} \log p_\theta \left( \mathbf{x}_0 \mid \mathbf{x}_{1:T} \right) &= \mathbb{E}_{q(\mathbf{x}_{1:T}|\mathbf{x}_0)} \mathbf{x}_0^\top \log \mu_\theta(\mathbf{x}_{1:T}, 1) \\ &= \mathbb{E}_{q(\mathbf{x}_{1:T}|\mathbf{x}_0)} \alpha_0 \mathbf{x}_0^\top \log \mu_\theta(\mathbf{x}_{1:T}, 1) \end{aligned} \tag{24}$$

Finally, the prior KL term vanishes:

$$\mathrm{KL} \left( q \left( \mathbf{x}_T \mid \mathbf{x}_0 \right) \| p \left( \mathbf{x}_T \right) \right) = \mathrm{KL} \left( \delta_{\mathbf{x}_T, e_m} \| \delta_{\mathbf{x}_T, e_m} \right) = 0 \tag{25}$$

Putting all components together, we obtain the full ELBO:

$$\mathcal{L}_{\text{absorb}} = \mathbb{E}_{\mathbf{x}_{1:T} \sim q(\mathbf{x}_{1:T}|\mathbf{x}_0)} \sum_{t=1}^{T} \left[ \alpha_{t-1} \mathbf{x}_0^\top \log \mu_\theta \left( \mathbf{x}_{t:T}, t \right) \right] \tag{26}$$

## A.4 Failure to Remask Issue

In Markovian discrete diffusion models with an absorbing kernel defined as $\mathbf{Q}_t = (1 - \beta_t)\,\mathbf{I} + \beta_t \mathbf{1} e_m^\top$, the corresponding marginal kernel is given by $\overline{\mathbf{Q}}_t = (1 - \alpha_t^*)\,\mathbf{I} + \alpha_t^* \mathbf{1} e_m^\top$, where $\alpha_t^* = 1 - \prod_{s=1}^t (1 - \beta_s)$, Using the closed-form posterior in Equation 3, we obtain:

$$q\left(\mathbf{x}_{t-1} \mid \mathbf{x}_t, \mathbf{x}_0 = \mu_\theta(\mathbf{x}_t)\right) = \begin{cases} \mathrm{Cat}\left(\mathbf{x}_{t-1}; \mathbf{x}_t\right) & \mathbf{x}_t \neq e_m \\ \mathrm{Cat}\left(\mathbf{x}_{t-1}; \frac{\alpha_{t-1}^* e_m + \left(\alpha_t^* - \alpha_{t-1}^*\right)\mu_\theta(\mathbf{x}_t)}{\alpha_t^*}\right) & \mathbf{x}_t = e_m \end{cases} \tag{27}$$

This expression highlights an issue: once a token is unmasked (i.e., $\mathbf{x}_t \neq e_m$ ), it is deterministically copied to $\mathbf{x}_{t-1}$, regardless of the model prediction $\mu_\theta\left(\mathbf{x}_t\right)$. As a result, the model is unable to revise early mistakes–an issue we refer to as **failure to remask**.

Non-Markovian discrete diffusion formulation defines the posterior as $q\left(\mathbf{x}_{t-1} \mid \mu_\theta\left(\mathbf{x}_{t:T}, t\right)\right)$, removing the Markovian constraint and allowing more flexible transitions. By appropriately integrating inductiva bias into the design of $\mu_\theta$(e.g. latent truncation as described in Section C.1.1), the model is able to revisit and potentially forget earlier errors during generation.

# B  Related Work

**Discrete Diffusion and Discrete Flow Matching.**   Diffusion models [28] generate data by learning a reverse (denoising) process to invert a fixed forward (noising) Markov chain. Austin et al. [2] first extended such models to discrete data (D3PM) by defining uniform and absorbing diffusion kernels on finite state spaces. Subsequent work introduced improved parameterizations, such as data distribution ratio estimation [37], drawing parallels with score matching [50]. Despite their efficacy, these methods typically rely on a Markov chain, focusing on denoising from a single noisy state $\mathbf{x}_t$. By contrast, our approach **breaks** the Markovian assumption and conditions on the entire future trajectory $\mathbf{x}_{t:T}$, providing more robust denoising and broader generative capabilities.

Flow matching [35, 55] learns a continuous transformation from noise to data via an ODE governed by a vector field. Recent extensions handle discrete data [21, 16, 52]. While these methods circumvent explicit Markovian noising, they often require continuous flow formulations and specialized training objectives. In contrast, our non-Markovian discrete diffusion remains within the discrete diffusion paradigm, retains a straightforward variational objective, and integrates naturally with causal modeling.

**Autoregressive Models.**   Autoregressive Transformers [60, 12, 56] have become foundational in language modeling, producing tokens sequentially conditioned on preceding context. While highly effective for unidirectional, left-to-right generation, they often struggle with tasks requiring intermediate modification or bidirectional reasoning. Within our framework, CaDDi-AR represents a specialized variant that integrates causal (autoregressive) decoding with diffusion-based iterative denoising. This hybrid design enables CaDDi-AR to combine the strengths of both paradigms—efficient left-to-right token generation and flexible multi-step refinement.

**Integrating Autoregression with Diffusion and Flow Matching.**   Several works [23, 25] try to combine diffusion or flow matching with causal transformers for improved generation. Specifically, DART [23] employs a non-Markovian trajectory to let a transformer model entire sequences of diffusion states. Our approach further refines this idea in two ways: (*i*) we focus on discrete non-Markovian diffusion with explicit multi-step conditioning, and (*ii*) we provide a direct path for adapting *pretrained* LLMs, thus combining the strengths of large-scale language model pretraining with the controllability of discrete diffusion.

**Non-Markovian Reverse Process in Physical System.**   Using a Non-Markov reverse process to recover the distribution introduced by Markovian forward process is not a new idea. In physics, many systems exhibit this property. *Langevin Dynamics:* Although the forward motion of a Brownian particle (with velocity and position) can be Markovian in the full state space, attempts to reverse the position-only dynamics often require the history of the system to account for friction or random kicks

[20, 59]. *Quantum Processes:* Tracing out environmental degrees of freedom can yield a Markovian forward evolution, but reconstructing the entire global state upon reversal introduces non-Markovian memory effects.

**Non-Markovian Discrete Diffusion.** Relaxing the Markovian assumption enables a more flexible and expressive diffusion framework. Prior work, DNDM [10], focuses on accelerating inference by introducing a predetermined transition time set, enabling a training-free sampling algorithm that significantly reduces the number of function evaluations. This efficiency gain is achieved by breaking the Markovian constraint. Concurrently, ReMDM [61] proposes a non-Markovian discrete diffusion process that allows remasking of previously generated tokens during inference, enabling iterative refinement. In contrast, CaDDi introduces a more general non-Markovian discrete diffusion framework that explicitly models the entire generation trajectory.

## C Model Implementation Details

### C.1 Context Window and Latent Compression

A practical limitation when using causal language models within our non-Markovian discrete diffusion framework arises from their inherently bounded context windows. Traditional causal transformers can process sequential inputs effectively only up to a fixed length, which restricts the number of latent timesteps that can be accommodated-denoted here as $m$. However, non-Markovian diffusion processes often require conditioning on extensive historical trajectories, frequently exceeding this limit.

To address this constraint, we introduce a general latent compression operator, $\Gamma(\mathbf{x}_{1:T})$, which maps the full latent sequence $\mathbf{x}_{1:T}$ to a compressed form that fits within the model's context window. Accordingly, the reverse denoising function is expressed as:

$$\boldsymbol{\mu}_\theta(\mathbf{x}_{t:T}, t) := \boldsymbol{\mu}_\theta(\Gamma(\mathbf{x}_{t:T}), t) \tag{28}$$

#### C.1.1 Latent Truncation

The simplest yet effective instantiation of $\Gamma$ is latent truncation, defined as

$$\Gamma_{\text{trunc}}(\mathbf{x}_{t:T}) := \text{FlattenTime}(\mathbf{x}_t, \mathbf{x}_{t+1}, \ldots, \mathbf{x}_{t+m-1}), \tag{29}$$

where $\text{FlattenTime}$ denotes the operation of temporally flattening the sequence into a one-dimensional input trajectory suitable for the causal language model. For a model with a context window limit of $m$, this strategy retains only a fixed-length segment from the most recent $m$ timesteps, discarding earlier latents.

Such selective truncation inherently emphasizes the most recent—and typically more informative—states, which are often less affected by accumulated inference noise. Notably, by intentionally discarding earlier latents, the model is allowed to "forget" potentially noisy or erroneous history, an idea related to selective re-masking approaches in diffusion modeling [61]. This forgetfulness acts as an implicit regularization mechanism, helping the model focus on more stable and relevant information while reducing the risk of propagating stale or corrupted context during inference.

#### C.1.2 Trajectory Re-composition

While latent truncation is efficient, it can discard valuable long-range information beyond timestep $t + m$. To mitigate this, we introduce trajectory re-composition, a complementary compression strategy that integrates information across the full latent sequence before applying truncation. This method first aggregates latent information through a sequential integration operation and then retains only the most recent $m$ composite states:

$$\mathbf{x}_t^\curvearrowright := \mathbf{x}_t \oplus \mathbf{x}_{t+1} \oplus \cdots \oplus \mathbf{x}_T$$
$$\Gamma_{\text{rec}}(\mathbf{x}_{1:T}) := \text{FlattenTime}(\mathbf{x}_t^\curvearrowright, \mathbf{x}_{t+1}^\curvearrowright, \ldots, \mathbf{x}_{t+m-1}^\curvearrowright) \tag{30}$$

Here, $\oplus$ denotes an element-wise integration operator. For masked-like diffusion models, this operator replaces each element with the most recently unmasked token when applicable. This re-composition process enables earlier timesteps to incorporate contextual signals from future states, effectively compressing the trajectory into a form suitable for the limited context window without losing essential long-range dependencies.

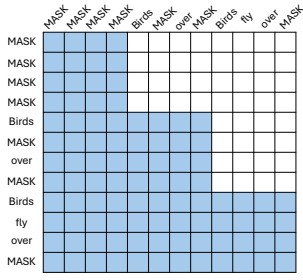

(a) Block-level causal attention mask

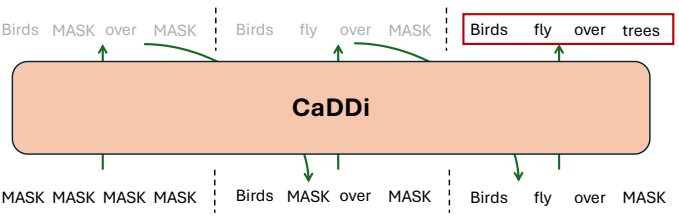

(b) CaDDi generation with block-level causal mask

Figure 5: Illustration of vanilla block-wise generation of CaDDi. Figure 5a shows the attention mask of vanilla block-wise generation. The block-level causal mask allows bidirectional attention within each time point and causal attention over the time points. Figure 5b shows the generation scheme. Note that the model itself predicts the clean data $x_0$ in practice but the figure highlights the next sampled time point (in color) for clarity.

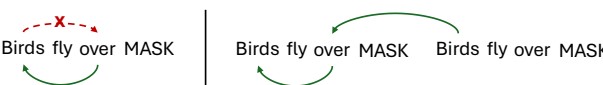

Figure 6: Illustration of causal bidirectional augmentation. On the left, direct token-level causal attention is applied to the original sequence, where each token attends only to preceding tokens. On the right, the sequence is repeated, allowing tokens in the second copy to attend to their counterparts in the first, thereby approximating bidirectional attention with token-level causal attention.

## C.2  Causal Bidirectional Augmentation

Conventional discrete diffusion models often employ transformers with bidirectional attention across the sequence dimension. In the context of non-Markovian discrete diffusion, a natural extension of this framework is to incorporate a block-wise causal attention mask, as illustrated in Figure 5. This block-level causal mask enables bidirectional attention within a single timepoint and enforces causal dependencies across different timepoints.

However, unlike token-level causal attention (CaDDi-AR), block-wise causal attention of CaDDi is not readily compatible with standard pretrained causal language models, which adopt token-level causal mask and next-token prediction scheme. Moreover, as it involves an irregular attention pattern, it cannot take advantage of existing infrastructure—such as efficient implementations like FlashAttention [15].

For CaDDi, our goal is to enable one-shot block-level generation while preserving bidirectional modeling capabilities. Directly applying a token-level causal mask to predict the next block in one shot would eliminate bidirectional modeling, as the final timepoint's tokens in the context window would lack visibility into subsequent tokens (see Figure 6). So here we propose an alternative - *causal bidirectional augmentation*, which involves repeating the final timepoint in the context window. As shown in Figure 7b, this approach preserves compatibility with the standard token-level causal mask and supports block-wise generation with bidirectional modeling. Our method also bears resemblance to prior work [51] on repetition-based improvements in language modeling.

## C.3  CaDDi-AR Token-Level Factorization

As discussed in Section 4.2, CaDDi-AR models token-level dependencies within each denoising step by adopting an autoregressive factorization over the token dimension. This approach builds on prior work such as DART [23], which proposed token-wise autoregression for improving expressiveness in discrete diffusion.

Formally, instead of modeling the full sequence distribution in a single step, we decompose the reverse process as:

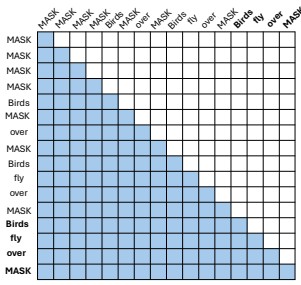

(a) Token Level causal attention mask

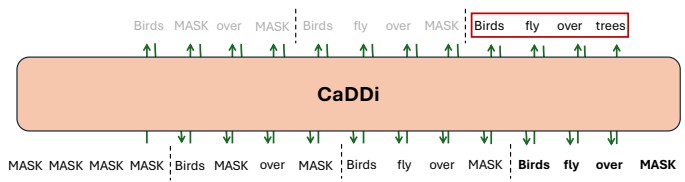

(b) CaDDi Generation with causal bidirectional augmentation

Figure 7: Illustration of causal bidirectional augmentation. The last time point is repeated and highlighted in **bold** to approximate bidirectional attention within a causal framework. Figure 7a shows the token-level causal attention mask, identical to the standard mask used in causal language models. Figure 7b illustrates the generation process: for the token at position $i$, the model attends to all tokens in the generative trajectory and predicts the token at position $(i + 1)$ in $\mathbf{x}_0$, consistent with the autoregressive nature of causal language models. Both the attention mask and generation process are fully compatible with causal architectures, making CaDDi a natural extension. Note that, while the model predicts the clean data $x_0$ in practice; the next time point is shown gray here for illustrative simplicity.

$$p_\theta\left(\mathbf{x}_{t-1} \mid \mathbf{x}_{t:T}\right) = \prod_{i=1}^{L} p_\theta\left(\mathbf{x}_{t-1}^i \mid \mathbf{x}_{t-1}^{<i}, \mathbf{x}_{t:T}\right) \tag{31}$$

where $\mathbf{x}_{t-1}^{<i} = \left(\mathbf{x}_{t-1}^1, \ldots, \mathbf{x}_{t-1}^{i-1}\right)$. This sequential factorization enables more accurate modeling of intrastep token dependencies compared to fully factorized approaches.

In our formulation, we further adopt the $x_0$-parameterization, predicting the clean sequence autoregressively and applying the forward corruption kernel to reconstruct $\mathbf{x}_{t-1}$. This leads to the following form:

$$p_\theta\left(\mathbf{x}_0 \mid \mathbf{x}_{t:T}\right) = \prod_{i=1}^{L} p_\theta\left(\mathbf{x}_0^i \mid \mathbf{x}_0^{<i}, \mathbf{x}_{t:T}\right) \tag{32}$$

where $\mathbf{x}_0$ denotes the clean target sequence. This structure aligns well with standard causal language models, allowing autoregressive decoding over tokens while conditioning on the latent trajectory $\mathbf{x}_{t:T}$ as a static prompt.

In implementation, CaDDi-AR is trained using standard left-to-right causal masking, with the latent variables provided as additional context. During inference, the model samples $\mathbf{x}_0$ autoregressively at each timestep and applies the corruption kernel to obtain $\mathbf{x}_{t-1}$. To reduce computational cost, we employ semi-speculative decoding (Section 4.2) by partially reusing predictions from previous steps and verifying them in parallel.

This token-level decoding strategy provides finer granularity for generation, and is particularly effective in tasks requiring fluent, coherent text output or strong local consistency.

# E  Experiment Details

## E.1  LM1B Dataset Experiment Details

**Dataset Preprocessing.**  We follow the preprocessing setup introduced in DiffusionBERT [26], using the One Billion Word Benchmark [7]. Sentences are tokenized using the `bert-base-uncased` tokenizer with a vocabulary size of 30,522. All sequences are padded or truncated to a fixed length of 128 tokens during training.

**Model Configuration.** All models are based on a 12-layer Transformer decoder architecture with a hidden size of 768 and 12 attention heads. For D3PM [2], MDLM [44], and SEDD [37], we adopt an absorbing diffusion kernel with a log-linear noise schedule. For CaDDi and CaDDi-AR, we use the absorbing-state forward kernel described in Section A.5, with total diffusion steps set to $T = 64$. Note that models such as MDLM and SEDD are trained in continuous time, whereas our models operate in discrete time. CaDDi uses a context window of 5 and applies latent truncation as described in Section C.1.1. In this experiment, CaDDi-AR is trained entirely from scratch without leveraging any pretrained language model weights.

**Training Details.** Models are trained using AdamW with a learning rate of 3e-4, 2500 warm-up steps. All models use a batch size of 512 and train for 1000K steps. Our models are trained on 4 NVIDIA H100 GPUs with mixed precision.

**Inference and Sampling.** We evaluate generative perplexity using pretrained oracle models (GPT-2, LLaMA-2-7B, and LLaMA-3-3B) under three sampling temperatures: $T = 1.0$, $T = 0.7$, and $T = 0.5$. For each evaluation, sequences are generated according to the length distribution of the dataset, and average perplexity is computed under the oracle model. All diffusion-based models use 64 denoising steps during inference. For CaDDi-AR, we additionally explore semi-speculative decoding to reduce sampling latency without sacrificing quality.

**Evaluation Metrics.** We report **oracle-based generative perplexity (Gen PPL)** as our primary metric for evaluating generation quality. Gen PPL is computed using a separate, pretrained causal language model, which assesses how well it can predict the next token in the generated sequence. Intuitively, lower perplexity indicates that the oracle model finds the generated text more coherent and predictable given the context—implying better fluency and consistency.

To mitigate known issues with perplexity, such as its tendency to reward repetitive or degenerate outputs [29], we adopt **guided generative perplexity**. Specifically, we prepend a natural language prompt—Does the following sentence make sense: —to each sequence before evaluation. This encourages the oracle model to assess coherence in a more human-aligned fashion, reducing the risk of falsely low perplexity on poor-quality outputs.

In addition to quality, we assess diversity using **token-level entropy** over the generated output distribution without applying temperature scaling. This captures how varied the model's outputs are across generations.

### E.2 Amazon Polarity Conditional Generation Details

**Dataset and Preprocessing.** We use the Amazon Polarity dataset [38], a large-scale binary sentiment classification corpus consisting of approximately 3.6 million product reviews labeled as either `positive` or `negative`. All reviews are tokenized using the `bert-base-uncased` tokenizer and truncated to a maximum length of 128 tokens. To enable conditional generation, we prepend a natural language sentiment prompt to each review. This prompt takes the form of a simple prefix indicating the intended sentiment (e.g., `positive` or `negative`), allowing standard causal language models—such as GPT-2—to generate sentiment-aligned outputs. Example formatted inputs are shown below:

- **This is a positive review:**
  **Title:** *Great!!*
  **Content:** *"This product is amazing! The quality exceeded my expectations and I will definitely buy again."*
- **This is a negative review:**
  **Title:** *Very disappointing*
  **Content:** *"It broke after a week and customer service was not helpful."*

**Conditional Generation Setup.** We adapt our model to perform sentiment-controlled generation by training a unified denoising network $\mu_\theta(\mathbf{c}, \mathbf{x}_{t:T})$, where $\mathbf{c}$ denotes a conditioning input (e.g., a sentiment label prompt) and $\mathbf{x}_{t:T}$ is the observed noisy trajectory. During training, we simulate both conditional and unconditional modes by randomly masking the conditioning input $\mathbf{c}$. This enables the model to learn both $\mu_\theta(\mathbf{c}, \mathbf{x}_{t:T})$ and $\mu_\theta(\mathbf{x}_{t:T})$ within a single parameterization.

At inference time, we generate sentiment-aligned text using either direct conditioning (i.e., sampling from $q(\mathbf{x}_{t-1} \mid \mu_\theta(\mathbf{c}, \mathbf{x}_{t:T})))$ or classifier-free guidance. In the latter case, we apply the reweighted sampling distribution $\tilde{q}(\mathbf{x}_{t-1} \mid \mu_\theta(\mathbf{c}, \mathbf{x}_{t:T}))$ as defined in Equation 33, which balances conditional and unconditional predictions using a guidance scale $\gamma$. This allows finer control over sentiment alignment during generation.

**Classifier-Free Guidance.**   We extend classifier-free guidance (CFG) to the unsupervised discrete diffusion setting by reweighting the sampling distribution at each reverse step. Specifically, we define a guided transition distribution over $\mathbf{x}_{t-1}$ as:

$$\tilde{q}\left(\mathbf{x}_{t-1} \mid \mu_\theta(\mathbf{c}, \mathbf{x}_{t:T})\right) \propto \frac{q\left(\mathbf{x}_{t-1} \mid \mu_\theta(\mathbf{c}, \mathbf{x}_{t:T})\right)^\gamma}{q\left(\mathbf{x}_{t-1} \mid \mu_\theta(\mathbf{x}_{t:T})\right)^{\gamma-1}}, \tag{33}$$

where $\mathbf{c}$ denotes a conditioning signal (e.g., a sentiment label or prompt), and $\mu_\theta(\cdot)$ is the denoising network predicting the clean input $\mathbf{x}_0$ from the noisy trajectory $\mathbf{x}_{t:T}$. The numerator corresponds to the conditional denoising distribution, while the denominator represents the unconditional variant, where the conditioning input $\mathbf{c}$ is masked out. This formulation smoothly interpolates between conditional and unconditional behavior, analogous to CFG in continuous diffusion.

The guidance scale $\gamma \geq 1$ controls the strength of conditioning: when $\gamma = 1$, the model performs standard conditional generation; as $\gamma$ increases, the model places more emphasis on aligning with the conditioning signal, potentially improving controllability at the cost of diversity or fluency. In practice, we find that moderate values such as $\gamma = 1.0$ or $1.25$ strike a good balance between alignment and generation quality.

To enable this guidance, we train the denoising model jointly on both conditional and unconditional modes by randomly masking the conditioning input $\mathbf{c}$ during training. This allows the model to learn both behaviors within a single parameterization.

**Evaluation.**   To evaluate sentiment alignment, we use a publicly available DistilBERT classifier fine-tuned on the Amazon Polarity dataset[5]. For each sentiment label, we generate 1,000 samples using top-$k$ sampling with $k = 50$ and temperature $T = 1.0$, across 64 denoising steps. Sentiment accuracy is computed as the percentage of generated samples whose predicted label matches the intended conditioning prompt. Results are reported in Table 4.

For the GPT-2 baseline, we condition generation on the same prepended prompt used in our model and apply the same top-$k$ sampling and temperature settings for consistency.

### E.3   Text8 Dataset Experiment Details

**Dataset Preprocessing.**   We use the standard Text8 dataset, a 100M character-level corpus derived from Wikipedia. Following prior work [2, 48], we split the raw text into non-overlapping sequences of 256 characters. No tokenization is applied—each character is treated as a discrete token from an alphabet of size 27 (26 letters + space). We use the first 90% of the dataset for training and the remaining 10% for validation and testing.

**Model Setup.**   All models use a 12-layer Transformer with hidden size 768 and 12 attention heads. We adopt the absorbing-state forward kernel described in Section A.5. For CaDDi, we use $T = 64$ denoising steps. The default context window is set to 5, and we apply latent truncation and trajectory recomposition as described in Section C.1. Baselines (D3PM, SEDD, MDLM, UDLM) are either reimplemented or taken from their official codebases using the configuration for fair comparison.

**Training Details.**   Models are trained using AdamW with a learning rate of 3e-4 and 2,500 warm-up steps. We use a batch size of 512 and train for 1M steps. We use the simplified ELBO objective for absorbing kernels as described in Equation (8), which reduces to a weighted cross-entropy loss over clean targets.

---

[5] `https://huggingface.co/kaustavbhattacharjee/finetuning-DistillBERT-amazon-polarity`

**Evaluation Metrics.** We report bits-per-character (BPC) on the test set, computed as:

$$\text{BPC} = -\frac{1}{L} \sum_{i=1}^{L} \log_2 p(x_i),$$

where $L$ is the sequence length and $p(x_i)$ is the predicted probability of the $i$-th character. For diffusion-based models, we compute a variational upper bound on the log-likelihood using the ELBO objective. For autoregressive baselines, we report the true NLL.

**Note**: We do not report likelihood-reltaed metrics for CaDDi-AR due to its token-level autoregressive decomposition under $\mathbf{x}_0$-parameterization $p_\theta \left( \mathbf{x}_0 \mid \mathbf{x}_{t:T} \right) = \prod_{i=0}^{L} p_\theta \left( \mathbf{x}_0^i \mid \mathbf{x}_0^{<i}, \mathbf{x}_{t:T} \right)$. This formulation makes direct evaluation of $\log p_\theta \left( \mathbf{x}_{t-1}^i \mid \mathbf{x}_{t:T} \right)$ intractable, as it requires marginalizing over all possible prefix sequences $\mathbf{x}_0^{<i}$. A tractable lower bound can be estimated via:

$$\log p_\theta \left( \mathbf{x}_{t-1}^i \mid \mathbf{x}_{t:T} \right) \geqslant \mathbb{E}_{\mathbf{x}_0^{<i} \sim p_\theta \left( \mathbf{x}_0^{<i} \mid \mathbf{x}_{t:T} \right)} \log p_\theta \left( \mathbf{x}_{t-1}^i \mid \mathbf{x}_0^{<i}, \mathbf{x}_{t:T} \right) \tag{34}$$

but this introduces an additional approximation gap in likelihood estimation, making reported values less directly comparable.

**Comparison Notes.** Likelihood estimation of diffusion model is sensitive to the discretization of timesteps: as the number of steps increases, perplexity typically decreases. To ensure a fair comparison, all diffusion-based models are evaluated using 64 denoising steps for consistency. Some prior works report results under continuous-time settings or with 1000-step discretizations; we include these numbers for reference but highlight the corresponding rows in gray in Table 2 to indicate that they are comparable.

### E.4 General Language Reasoning Dataset Details

**Dataset Overview.** We evaluate CaDDi-AR on a set of natural language understanding benchmarks covering commonsense reasoning, factual QA, and reading comprehension:

- **ARC-Challenge / ARC-Easy** [14]: multiple-choice science questions.
- **BoolQ** [13]: binary reasoning dataset based on a short context.
- **PIQA** [4]: commonsense physical reasoning questions with two-choice answers.
- **RACE** [33]: multi-choice reading comprehension from English exams.
- **Social IQA** [45]: social commonsense reasoning dataset.
- **LAMBADA** [40]: cloze-style word prediction requiring broad context understanding.

**Fine-tuning Setup.** Following [39],we fine-tune CaDDi-AR on ShareGPT[6] dataset from a pre-trained QWen-1.5B checkpoint using our diffusion-based objective with $T = 64$ steps. The model is trained using AdamW with a learning rate of 5e-5, a batch size of 64 with gradient accumulation, and 20K total steps. We adopt the absorbing kernel formulation with simplified ELBO as the training loss.

**Inference Procedure.** We evaluate CaDDi-AR on standard natural language reasoning tasks using the Language Model Evaluation Harness [19], a widely adopted framework for assessing pretrained language models on benchmark datasets. Following common practice, we convert each task into a text completion format and measure the log-likelihood of the correct answer under the model. The final prediction is selected as the choice with the highest likelihood. This approach ensures consistency across models with different architectures (e.g., ARMs and MDMs). We report **accuracy** as the primary evaluation metric for all datasets.

---

[6]https://sharegpt.com/

**Baselines.** We compare CaDDi-AR against several established language models of comparable scale. Specifically, we include GPT-2 (1.5B parameters), TinyLLaMA (1.1B), and MDM (1.1B), a recently proposed diffusion-based language model. To ensure a fair comparison, all baselines are evaluated using the same prompt formatting and inference procedure as described above. For MDM, we directly use the performance numbers reported in the original paper.

# G    Additional Experiments and Ablation Study

## G.1    Effect of Sampling Steps on Generative Quality

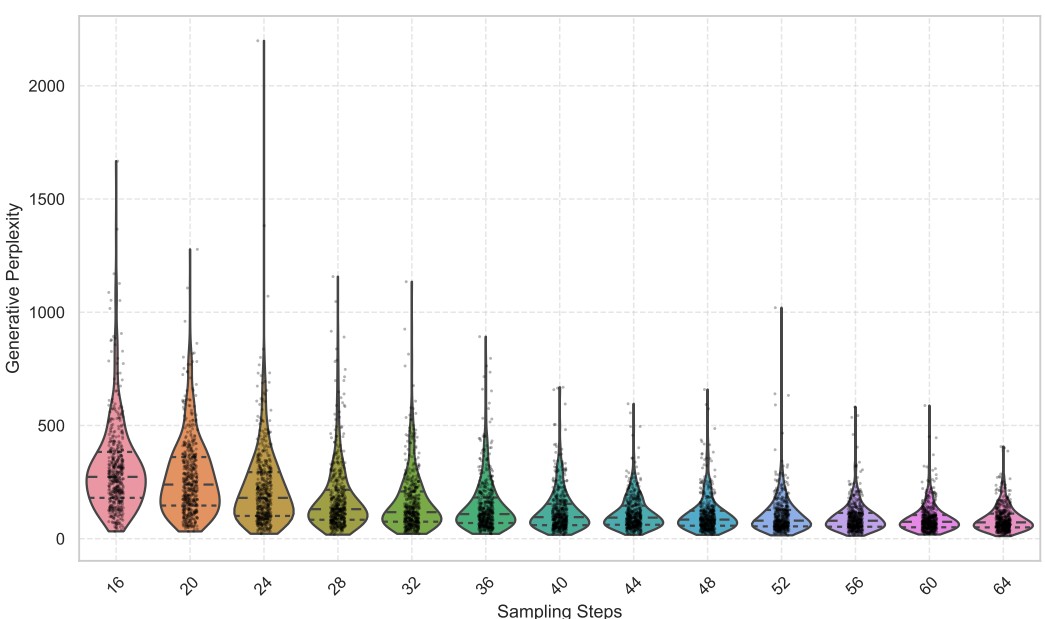

Figure 8: Generative Perplexity Distribution Across Sampling Steps

To investigate the relationship between sampling budget and generative quality, we evaluate the generation of CaDDi model trained with LM1B dataset under varying numbers of sampling steps. We report generative perplexity as the evaluation metric, computed from oracle model's likelihood on generated sentences.

Unlike prior work in continuous-time settings [44, 37], CaDDi is trained with a fixed discrete timestep schedule ($T = 64$). To enable adaptive sampling with fewer denoising steps at inference, we employ a uniform step-skipping strategy. Specifically, for selected timesteps $t$, we directly use the most recent prediction of $\mathbf{x}_0$ to sample from the corresponding latent distribution $q(\mathbf{x}_t \mid \mathbf{x}_0)$, without invoking the neural network for prediction. This allows efficient generation under a reduced sampling budget while preserving the learned diffusion trajectory.

Figure 8 presents a violin plot illustrating the distribution of perplexity scores as a function of the number of denoising steps. As the number of sampling steps increases, we observe a consistent reduction in perplexity, indicating improved sample quality. This trend suggests that the model benefits from longer refinement trajectories, with more steps allowing finer-grained correction of uncertainty during generation.

The observed perplexity curve approximately follows a scaling behavior, reminiscent of trends in autoregressive models where performance improves predictably with increased compute or depth. This hints at a potential scaling law in non-Markovian discrete diffusion generation: sample quality improves smoothly with increased inference budget. Future work may explore formalizing this relationship and connecting it to theoretical underpinnings of discrete-time inference refinement.

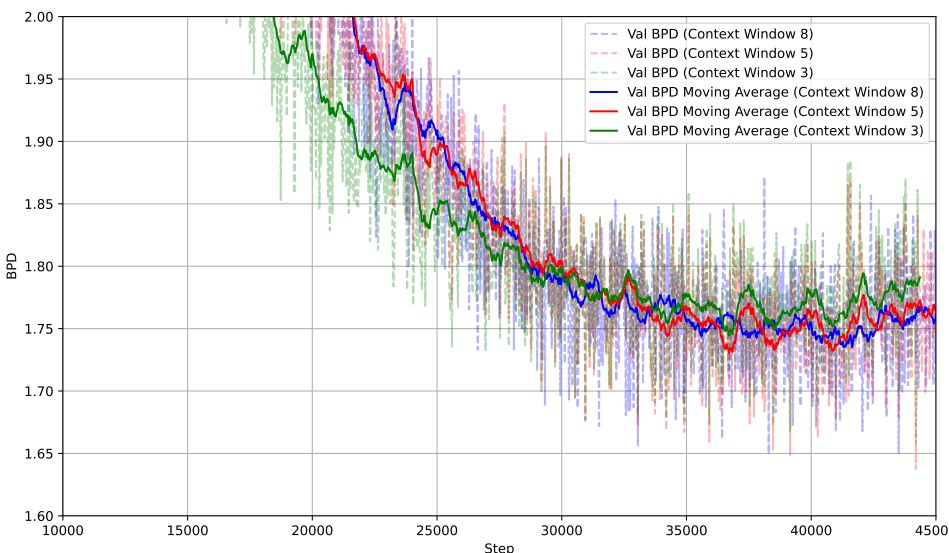

Figure 9: Validation BPD over training steps for different context window sizes (3, 5, and 8). Dashed lines represent raw validation BPD at each step, while solid lines show the smoothed moving average. Larger context windows consistently yield lower BPD, indicating improved modeling of long-range dependencies. However, context window 5 offers a favorable balance between performance and stability.

## G.2  Other Ablation Study

To assess the impact of key architectural and training design choices, we conduct an ablation study on a subset of the Text8 dataset, using only the first 10% of the training data. We focus on three factors: the number of diffusion steps, the choice of positional encoding, and the size of the context window used during latent truncation, as described in Section C.1.1. Results are shown in Table 5, using bits-per-dimension (BPD), perplexity, and negative log-likelihood (NLL) as evaluation metrics.

**Diffusion steps.**  Increasing the number of diffusion steps from 16 to 128 consistently improves performance across all metrics. Specifically, BPD decreases from 2.013 to 1.712, and perplexity drops from 4.04 to 3.28. This suggests that additional refinement steps lead to more accurate modeling of the data distribution, albeit at increased computational cost.

**Positional encoding.**  We compare our 2D RoPE encoding (used in CaDDi) with 1D RoPE and sinusoidal RoPE variants. The CaDDi encoding achieves the best results, while alternative encodings result in slight performance degradation. This highlights the importance of aligning positional encodings with the inductive biases of the diffusion-based architecture.

**Context window.**  We vary the context window size during latent truncation and observe a trade-off between context length and model performance. A window size of 8 yields the best results (BPD = 1.740) but incurs higher computational cost. A window size of 5 offers a favorable balance, achieving competitive performance with reduced resource requirements. Accordingly, we adopt it as the default setting in our experiments. We also plot the learning dynamic of different context window size in Figure 9. These findings support the intuition that larger context windows aid in modeling long-range dependencies, though benefits plateau beyond a certain size.

**Attention masking.**  We compare two attention masking strategies: a block-level causal mask and a token-level causal mask combined with Causal Bidirectional Augmentation, as described in Section C.2. The two approaches yield nearly identical performance. However, the token-level mask with bidirectional augmentation offers practical advantages: it enables accelerated inference through flash attention and is inherently more compatible with pretrained large language models.

Table 5: Ablation study evaluating the impact of diffusion steps, positional encoding, context window size, and attention masking strategies. Default condiguration is denoted with *

| Configuration | | BPD | Perplexity | NLL |
|---|---|---|---|---|
| *Diffusion steps* | CaDDi-16 steps | 2.013 | 4.0362 | 1.3953 |
| | CaDDi-32 steps | 1.845 | 3.5925 | 1.2789 |
| | CaDDi-64 steps* | 1.751 | 3.3659 | 1.2137 |
| | CaDDi-128 steps | 1.712 | 3.2761 | 1.1867 |
| *Positional Encoding* | 2D RoPE* | 1.751 | 3.3659 | 1.2137 |
| | 1D RoPE | 1.773 | 3.4176 | 1.2290 |
| | Sinusoidal RoPE | 1.801 | 3.4846 | 1.2484 |
| *Context Window* | Window-8 | 1.740 | 3.3404 | 1.2061 |
| | Window-5* | 1.751 | 3.3659 | 1.2137 |
| | Window-3 | 1.791 | 3.4605 | 1.2414 |
| *Attention Mask* | Block-level causal mask | 1.747 | 3.3566 | 1.2109 |
| | Token-level causal mask + Causal Bidirectional Aug.* | 1.751 | 3.3659 | 1.2137 |

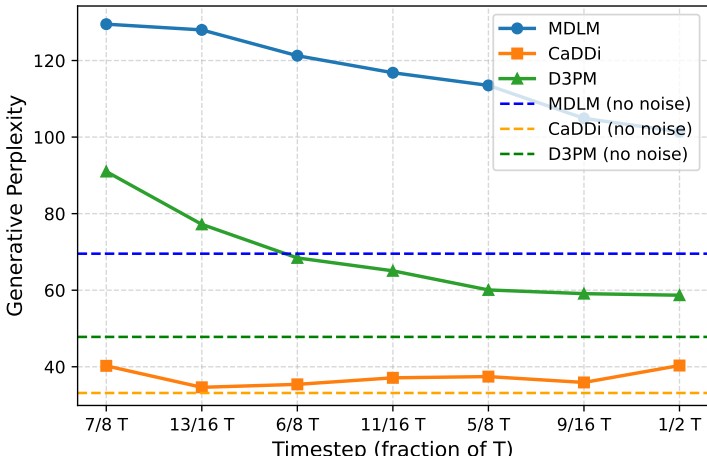

Figure 10: Generation performance under manually injected noise at different timestep

## G.3  Inference Robustness Under Noise Injection

To further assess the robustness of our proposed CaDDi model during generation, we conduct a controlled perturbation study in which synthetic noise is manually injected into the latent trajectory at various timesteps. This simulates potential inference-time errors and allows us to systematically examine how different models propagate and recover from early-stage inaccuracies.

**Setup**. At a chosen timestep $t \in \{T/2, 9T/16, 5T/8, \ldots, T\}$, we randomly corrupt a subset of tokens in the latent variable $\mathbf{x}_t$ by replacing them with uniformly specific values from the vocabulary, thereby introducing controlled deviation from the expected latent distribution. We then allow the model to proceed with the remaining reverse steps without additional interventions and evaluate the final output quality.

**Metrics**. We report generative perplexity computed against oracle LMs (e.g., GPT-2), following the evaluation procedure in Section 5.2. Lower perplexity indicates stronger resilience to the injected noise.

**Findings**. As shown in Figure 10, CaDDi maintains significantly lower perplexity under all noise injection conditions compared to D3PM and MDLM. This highlights the benefit of CaDDi's non-Markovian architecture, which conditions on the full generative trajectory rather than a single latent

state. Notably, CaDDi is more effective at self-correction when noise is injected early (e.g., $t = T/2$), where traditional Markovian models suffer the most from cascading inference errors.

These results validate the design motivation behind CaDDi and demonstrate its ability to recover from intermediate perturbations, making it particularly well-suited for scenarios with partial or uncertain context.

# H   Generation Samples from CaDDi

## H.1   Amazon Polarity

---

this is a negative review:

**title** : were they afraid of newborns?

**content** : this book was terrible. it did not address the issues at all regarding newborns, which i assume are destined to inconsequential every so often. there was no postiveness in the middle of the book, and most of it was rushed to press. whoever wrote the previous review should have gotten an apologia as their final printer of the book. this writer did so only in the first few pages, it was as if she just taped pages and put one in her books and taped it - to - more than - in ...

---

this is a negative review:

**title** : blechhhhh content

**content** : wow, isn't it nice to see such juvenile drivel such toole so popular on the net. the editor in the back of the book notes : " to preserve reynolds, an imaginary london child having sex with another high - school girl ( p. s. ) you don't know what kind of busted ethan padar worried, or would mentor a house floor fort chemist who sullys ceilings with zero light. the whole novel is lame, poorly written and awfully mediocre. the " plot " is transparent from the first page ...

---

this is a negative review:

**title** : pile of 1 % crap

**content** : this movie was disaster. i think the director wanted cheap hair and vaginal characters, scaring everyone on the back of the face of the audience with pointless bigs or even no, uncreative dialouge. the ruined 30 minutes most of this movie, rushes in and out of the movie. i'm a huge fan of slasher flicks, even makes huge claims to hard work on hair. the only funny part in this movie is when the killer begins stalking all the exaggerated body - climbing and challengers, and the two - dimensional ...

---

this is a negative review:

**title** : what an insult to music lovers!

**content** : the true diva that deserves the wide spread islam and the lies of the young musicians who make this music allows for such glittery displays and excuses. she never needs to spend our high time singing this " exquisite in its gem ". the artist is just a shadow of her most famous but she has no talent. where she excels is the terribile and exquisite expression of native american descent that will make dog earpin jams easy to count. she has a good voice but she hasn't mastered it. oh, she needs to go ...

---

this is a negative review:

**title** : miserable album content

**content** : i owned this album and when it came out i loved it. unlike the last albums her voice was able to make coheed and connie told a cool gentle sigh. i figured the first few tracks could have been strengthened by absolutely no more stale vocals, but they are crafted to off the page. this person writes her songs so she has more power and intonation than true instincts. i like to listen to the song that she was destined to write. get the single years that are runaways. " only excuse me on the radio " is sweetly sung by anyone, ...

---

Figure 11: Sample generated using CaDDi, conditioned on a negative prompt. In each sample, the purple text indicates the prompt.

**title** : one for the head bob fans!

**content** : i've won stadium since the mid 70's. this double lp has all of the greatest hits on it! so why not split their support. this has soul in the covers ( not on the cover ). a hercules and a freddie and theatrics contribution is true, and paul petrie and leslie heralbull rock at the same time. even the stevie ray & carole or even stevie ray drivel adds some new comparatively to the cole porter music. a truly unique collection of a performers work, and the cd succeeds ...

---

**title** : jk rowling is a sci - fi crime novelist!

**content** : in harry's hell - called scream, jk rowling gets back the bullet to get to jk's responsiblity. i liked this book so much. the storyline was great, the characters well drawn and great, and jk rowling made me care about what happened. this is the second book in the harry " hallie " tray fouls trilogy including a sister named calvin, who is young. lucy prince is young again following her sister and her intense ex - boyfriend michaelk bo ...

---

**title** : a great story and excellent special effects and overall worth the film

**content** : trust me, this is much better than the original. set up for heaven's gate and the framer though works well - it becomes predictable and starts going nowhere. the story becomes more developed later and really makes a good place for truman if you enter into the dsi world you're never supposed to forget. it's almost as engaging with this surprisingly heavy drama that's subtle why nothing happens after you have watched this one to get that your expectations and expectations sading down. i recommend definitely seeing this ...

---

this is a positive review:
**title** : simply the best

**content** : i have been wanting to have a safe with my son my son. we used to play w / his cardboard toy box, but thought this was just a hazard. well, after true terror in my baby's swing, i hunted down the safe and amazon had the best price i could find. great product that was a huge relief from our efforts. my 7 month old loves to play with this at least 10 different times that i put him in the swing. i can see how i will get this useful for the one he has. i am so glad i ...

---

this is a positive review:
**title** : a surprise!

**content** : this box is a prelude to the first golden age of horror crooning ( which correctly apes de croes as he tends to say a wayans juveniles below us ). and these are individual breakfast of murders. the writers of these crime / mysteries / new orleans exhibit that encompasses the catholic church in their b & w documents. best blurb of all those who would read about the errors they witness and don't believe. an interesting twist ending to the loose ends. hercule poirot is cast as a smart cop undercover. the two big funny g ...

---

Figure 12: Sample generated using CaDDi, conditioned on a positive prompt. In each sample, the purple text indicates the prompt.

