# OpenReview forum: "Non-Markovian Discrete Diffusion with Causal Language Models"
_NeurIPS.cc/2025/Conference — NeurIPS 2025 poster_

### Official Review · Reviewer_WEg3 · 2025-06-23

**Clarity:** 3
**Significance:** 3
**Originality:** 1
**Rating:** 5
**Confidence:** 3

**Summary:**

This paper proposes a reformulation of discrete diffusion by relaxing the standard Markovian assumption. Specifically, the noising distribution assumes conditional independence given $x_0$ that is $p(x_{t} \mid x_{1:t-1}, x_0) = p(x_{t} \mid x_0)$ instead of $ p(x_t \mid x_{t-1})$ in standard diffusion, thereby removing the dependence on the previous noisy state.

The work begins by highlighting a key limitation of standard discrete diffusion models: the accumulation of decoding errors due to the strong dependency on previous state in the denoising step $p(x_{t-1} \mid x_t)$. This dependency exposes the inference procedure to compounding deviations from small prediction errors at each timestep.

To address this, the authors propose a non-Markovian approach in which the noise at each timestep is conditionally independent given the clean data $x_0 $. That is, each noisy state depends solely on time and the original input, eliminating dependence on previous noisy states.

Building on this conceptual framework, the paper introduces the Causal Discrete Diffusion Model (CaDDi). This model incorporates 2D Rotary Positional Encoding to handle the sequential and temporal dimensions inherent to discrete sequences. Furthermore, the authors propose an autoregressive variant, CaDDi-AR, which factorizes the joint distribution across the sequence. To mitigate the associated computational overhead, they introduce a semi-speculative decoding strategy.

The method is empirically validated through experiments, demonstrating its effectiveness.

**Questions:**

## Suggestions for Improvement

1. The authors should clearly articulate the novel contributions of this work compared to [1], especially given the close conceptual overlap.

2. A thorough literature review is necessary. Related works on compounding denoising errors and advanced discrete diffusion strategies should be discussed in one or more dedicated sections.

3. An empirical analysis of the computational cost (including the effect of semi-speculative decoding) would make the evaluation more rigorous and transparent.

4. A detailed explanation or formal proof of the first equality in Equation (6) would help ensure the mathematical soundness of the method.

## Question:

What is the benefit of conditioning on the whole sequence rather than on $x_t$?

**Ethical Concerns:**

["NO or VERY MINOR ethics concerns only"]

**Final Justification:**

1 & 2.
My initial evaluation was based on the assumption that the proposed model resembled existing non-Markovian discrete diffusion models. However, these models are non-Markovian only in the forward process. The authors have committed to clarifying this distinction and adding relevant references. Given this, the method presents a novel contribution to discrete diffusion and is a meaningful contribution to the field.

3 & 5.
The authors have provided a theoretical justification for the benefits of the method in the rebuttal. I encourage the author including elements of this discussion into the main paper. It would strengthen the core arguments and improve clarity.

4.
The computational cost is a recognized limitation, which the authors address by restricting the context window. Since this trade-off is central to the contribution, the empirical analysis would benefit from evaluating smaller window sizes. New experiment were not realistic in the tight rebuttal schedule, but the revised version would benefit of these additional experiments.

6.
The authors have committed to including a section on the societal impacts of their work.

**Limitations:**

## Limitations

- The paper does not discuss (or evaluate) the residual computational overhead resulting from the computation of the whole sequence $x_{t:T}$ during training.
- The paper does not discuss (or evaluate) the residual computational overhead after applying the semi-speculative decoding strategy.
- The paper does **not** include a discussion of potential negative societal impacts.

**Paper Formatting Concerns:**

No major formatting issues

**Quality:**

2

**Strengths And Weaknesses:**

## Strengths

- The paper is clearly written and easy to follow.
- The core methodology is intersting and sound.
- Beyond the central contribution, the paper introduces secondary innovations, including 2D rotary positional encoding, semi-speculative decoding, and leveraging pre-trained language models.
- The experiments demonstrate some benefits of the proposed approach.

---

## Weaknesses

### 1. Novelty

The main conceptual claim, namely, that compounding denoising errors in discrete diffusion can be alleviated via a non-Markovian assumption, has been previously introduced in [1]. While [1] focuses on graph generation, the underlying idea and motivation closely align with the present work. Therefore, the novelty of the contribution is limited.

### 2. Positioning and Related Work

The paper does not adequately position itself within the existing literature. In particular:

- The concept of compounding denoising errors is well-known and explicitly discussed in [2].
- Several recent works [3–7] propose methods to mitigate such errors through informed or planned denoising and sampling strategies.
- The relationship between the proposed semi-speculative decoding and prior decoding strategies (e.g., planning, correction, or speculative sampling [1, 3-7]) is not discussed. It remains unclear how the proposed strategy differs from or improves upon these approaches.

### 3. Computational Efficiency and Evaluation

The proposed method introduces two components that increase computational cost: conditioning on the full sequence and the use of an autoregressive (AR) sampling strategy.

1. The decoder conditions on the entire sequence $x_{t:T}$, which requires computing the full sequence during training. This introduces additional computational overhead compared to standard discrete diffusion conditioning only on $x_t$. However, this overhead is neither discussed nor evaluated in the paper. Furthermore, it is not evident that conditioning on the full sequence provides substantial benefit over conditioning only on $x_t$. An ablation study would be valuable to assess the impact of this design choice.

2. The authors acknowledge the computational cost of their AR method resulting from the high number of function evaluations (NFE) needed. There is however no quantitative evaluation of this cost, nor of the extent to which the proposed semi-speculative decoding mitigates it. Moreover, the experiments do not disentangle the benefits of the method itself from those attributable to increased computational budget. A more detailed analysis is necessary to clarify how much of the observed performance gain results from the method rather than from additional compute.


### 4. Theoretical Clarity

The first equality in Equation (6) is not self-evident. Providing a derivation or proof would be useful

---
[1] Boget, Y. *Simple and Critical Iterative Denoising: A Recasting of Discrete Diffusion in Graph Generation*. ICML 2025. https://openreview.net/forum?id=kPTW6hGrJy
[2] Lezama, J., et al. *Discrete predictor-corrector diffusion models for image synthesis*. ICLR 2023.
[3] Liu, S., et al. *Think while you generate: Discrete diffusion with planned denoising*. ICLR 2025.
[4] Zhao, Y., et al. *Informed correctors for discrete diffusion models*. https://arxiv.org/abs/2407.21243
[5] Kim, J., et al. *Train for the worst, plan for the best: Understanding token ordering in masked diffusions*. https://arxiv.org/abs/2502.06768
[6] Peng, F., et al. *Path planning for masked diffusion model sampling*. https://arxiv.org/abs/2502.03540
[7] Nie, S., et al. *Large language diffusion models*. https://arxiv.org/abs/2502.09992

---

> ### Author Rebuttal · Authors · 2025-07-31
>
> We thank the reviewer for their thoughtful and constructive feedback. We are pleased that the reviewer found our core methodology interesting and sound, and appreciated our secondary innovations, including 2D rotary positional encoding, semi-speculative decoding, and leveraging pre-trained language models. We summarize and address each of the raised weaknesses/limitation below.
>
> **1. Novelty and comparison with [1]**
>
> We appreciate the reviewer bringing [1] to our attention. While both works share the term "non-Markovian", we wish to emphasize that the two are conceptually distinct frameworks and that our contributions are novel and independent, and should not be overshadowed due to similarities merely in terminology. Key distinctions and novelties of our approach compared to [1] and other related works include:
> - **Reverse Process Formulation**: The key difference lies in how the reverse process is formulated. While [1] constructs non-Markovian forward kernels as $q(x_t | x_{t-1}, x_0)=q(x_t | x_0)$, their **reverse kernel $p_\theta(x_{t-1} \mid x_t)$ remains "Markovian"**. In contrast, a core insight of CaDDi is that under the assumption of an independent non-Markovian forward process, incorporating history into the reverse process $p_\theta(x_{t-1} \mid x_{t:T})$ yields a strict mutual information gain $I(x_{t-1};x_{t+1:T}|x_t) > 0$. This result highlights that conditioning on the generative trajectory strictly reduces the uncertainty about $x_{t−1}$ compared to conditioning on $x_t$ alone. As a result, only coupling the non-Markovian forward process with a non-Markovian reverse process can fully leverage the structure of the data. CaDDi is explicitly designed with a non-Markovian reverse process, which enables it to consistently outperform baselines across a wide range of experiments. This architectural and theoretical distinction is central to our contribution and sets our framework apart from [1].
> - **Unification with Causal Language Models**: Another central and novel contribution of CaDDi is its **unification of sequential (causal) and temporal (diffusion) reasoning within a single non-Markovian transformer architecture**. We demonstrate that CaDDi-AR with 2D Rotary Positional Encoding treats standard causal language models as a special case (T=1), enabling the direct reuse of pretrained LLM weights with no architectural changes. This is a practical advantage that allows us to **leverage the vast knowledge encoded in large-scale pretrained LLMs for discrete diffusion as shown in experiments in Table 3**. This specific unification and the ability to fine-tune pretrained LLMs for non-Markovian discrete diffusion is not explored in [1].
>
> PS: We also note that the simple iterative denoising in [1] can also be considered a special case of ReMDM (where all tokens are remasked in the subsequent step). We have already detailed the key differences between our approach and the more general ReMDM work in Appendix Section B. We appreciate this valuable feedback and will further expand it with [1] and clarify this discussion in the revised manuscript to explicitly highlight these distinctions.
>
>
> **2. Clarification on Positioning and Related Work**
> - **Compounding Denoising Errors Issue**: We acknowledge compounding denoising errors as a known issue in discrete diffusion, discussed in prior works like [2]. Our paper mentions this in the Introduction (Page 1, Section 1, paragraph 2) and Section 2 ("Error During Inference"). We will explicitly cite [2] and other relevant works. Our contribution is not identifying the problem, but proposing a fundamentally different architectural solution (non-Markovian conditioning on the full trajectory) to mitigate it, unlike predictor-corrector or sampling strategies.
>
> - **Other Mitigation Strategies [3-7]**: Thanks for referring to these work, some of them are already cited in Appendix Section B. While these works also aim to alleviate the compounding error problem, we notice that **all of them focus on inference techniques, which are orthogonal to our model's core contribution**. For instance, in [3], the improvement comes from an adaptive Gillespie algorithm replacing tau-leaping. [2,4] adopts additional corrector steps. In [5,6], they address the problem through additional planning (position choosing). Similarly, [7] uses a confidence score to bias the position for unmasking. All these methodologies are inference-time techniques. In contrast, our model uniquely defines a totally different non-Markovian discrete diffusion framework that inherently learns more robust dependencies across the  generative trajectory to solve the same problem. Crucially, **these techniques are orthogonal to our approach and can be combined with CaDDi**. For example, we can integrate low-confidence remasking from [7] and additional planning using Top-K probability margin from [5] into our CaDDi model, deciding which tokens need to be remasked again from the logits. Experiments have shown that these combinations can further benefit our model, proving their orthogonality as sampling tricks. **We thank the reviewer for raising these sampling techniques, as the suggestion helped improve the quality of our work**.
>
> | Model Configuration| GenPPL on LM1B↓ |
> | -| -|
> | CaDDi|142.51|
> | CaDDi + Low-Confidence Remasking [7]|98.81|
> |CaDDi + Top-K Probability Margin [5]|112.49|
>
> - **Relationship between Semi-Speculative Decoding and Prior Decoding Strategies**: It is important to emphasize that semi-speculative decoding is a speedup technique for AR-like models, specifically CaDDi-AR, without performance sacrifice. It is fundamentally different from the sampling/decoding techniques in [2-7], and thus not comparable. We will expand this explanation.
>
>
> **3. Clarification on the Benefit of Conditioning on the Whole Sequence Rather than just $x_t$**
> As discussed in point 1, our model uniquely conditions on the generative trajectory in the reverse process, thereby allowing the model to "revisit past states" and maintain a comprehensive view of the generation process, which significantly benefits performance. If the reviewer is referring to an ablation study concerning this non-Markovian reverse process, our Experiment Table 1 already provides this comparison, where MDLM can be considered a strong Markovian counterpart to CaDDi. CaDDi consistently outperforms MDLM across various metrics, demonstrating the benefits of our non-Markovian reverse process.
>
> Furthermore, if the reviewer is seeking an ablation study with models like [1], where the forward process is defined as non-Markovian but the reverse process remains Markovian (conditioning only on $x_t$), we have additionally run experiments to address this. Results with GPT2 generative perplexity show that conditioning on the generative trajectory in the reverse process offers more robust and effective inference:
>
>
> | Model | Reverse Process Condition | GenPPL on LM1B↓ |
> | -| -| -|
> | SID[1]|$x_t$-only|185.32|
> | CaDDi (Our Model)|trajectorty| 142.51|
>
>
>
> **4. Computational Cost and Additional Analysis**
>
> - **Overhead of Conditioning on Full Sequence**: The reviewer correctly identifies that conditioning on the entire sequence $x_{t:T}$ can introduce computational overhead. In Appendix Section C.1 ("Context Window and Latent Compression," Page 28), we discuss this practical limitation and introduce latent truncation and trajectory re-composition as strategies to compress the latent trajectory to fit within a bounded context window. Specifically, latent truncation (Section C.1.1) limits the conditioning to the most recent m timesteps, which directly controls the computational cost. We have already performed an ablation study on the context window size (Section G.2, Table 5, Figure 9), showing that a moderate window size offers a favorable balance between performance and computational cost.
> - **Quantitative Evaluation of Computational Cost**: Thanks for your suggestions! Below is a detailed quantitative evaluation of this cost on the LM1B dataset during generation. Speculative decoding is a dynamic speed-up technique, so the NFE has some randomness between different runs. As shown in the table below, semi-speculative decoding significantly reduces the NFE for CaDDi-AR while maintaining competitive performance, highlighting its effectiveness as a speedup technique. It's also worth noting that the non-AR version of CaDDi with 64 steps already achieves better performance than MDLM with 1000 NFE (and with larger FLOPs), which demonstrates that the improvement does not purely benefit from scaling up the computation, but also from the methodology (non-Markovian formulation, 2D ROPE, unification with LLMs).
>
> | Model | NFE | GenPPL on LM1B↓ |
> | -| -| -|
> |MDLM-64|64|199.45|
> |MDLM-500|500|188.93|
> |MDLM-1000|1000|179.41|
> |CaDDi|64|142.51|
> |CaDDi-AR|8192|137.82|
> |CaDDi-AR (with semi-sepculative decoding)|~656|139.80|
>
> **5. Theoretical Clarity about the first equality in Equation (6)**
> The first equality, $p_{\theta}(x_{t-1}|x_{t:T}):=q(x_{t-1}|x_{t:T}, x_0=\mu_{\theta}(x_{t:T},t))$, is a definition (as stated in line 68) based on the standard practice of parameterizing the reverse diffusion process using a prediction of the clean data $x_0$, which is widely used in prior work like D3PM, MDLM, ...In diffusion models, the true posterior $q(x_{t-1}|x_t,x_0)$ is often used as a target for the learned reverse process $p_{\theta}(x_{t-1}|x_t)$. In our non-Markovian setting, we extend this by conditioning on the future trajectory $x_{t:T}$.
>
> **6. Discussion of Potential Negative Societal Impacts:**
> We acknowledge the omission of a discussion on potential negative societal impacts and will add a dedicated section.
>
> Thanks again for your suggestions! We are confident that addressing these points will significantly strengthen our paper. We are committed to incorporating these revisions and providing any further clarifications needed.

---

> > ### Comment · Reviewer_WEg3 · 2025-08-02
> > **Concerns**
> >
> > Thank you for your response. I still have several concerns. Given the limited discussion time, I will focus on key issues:
> >
> > 5. The issue with Eq. 6 originates from Eq. 3. I do not believe the first equality represents D3PM; in D3PM, this is not a definition but a factorization.
> > $p(x_{t-1} | x_t) = \sum_{\hat{x}} p(x_{t-1} | x_t, \hat{x}) p_\theta(\hat{x} | x_t)$
> > Similarly, the first equality should be justified. Since Eq. 6 is central to the method, any incorrect formulation would be a serious concern for the paper.
> >
> > 3a. The core contribution of the method is the conditioning on the full denoising sequence, rather than only the last denoised step. A proper ablation is needed to isolate this effect. Table 5 only reports results for window sizes 3, 5, and 8. It should include at least size 1 (2 would also be informative). Moreover, the results do not clearly show that increasing window size improves performance. The gains appear marginal and are reported without error margins.
> >
> > 3b. The ability to revisit past states is not unique to full-sequence conditioning. Non-Markovian forward processes (e.g., [1]) can also enable this. Further justification is needed.
> >
> > 7 (New): My intuition is that conditioning on the full denoising sequence concentrates probability mass around distribution modes—akin to reducing the temperature parameter. This may not improve distribution modeling, but rather sharpen it. Could the authors comment?

---

> > > ### Author Response · Authors · 2025-08-03
> > >
> > > Thank you for your continued engagement with our work and for providing following-up reply. We will address each of your concerns as follows:
> > >
> > > **5. The Issue with Equation (6)**
> > >
> > > **Eq. 3 is exactly the same as D3PM**—they are simply written in different styles. Our Equation (3) $p\_\theta({x}\_{t-1} \mid {x}\_t)=q({x}\_{t-1} \mid {x}\_t, {x}\_0={\mu}\_\theta(\mathbf{x}\_t, t))$ represents a common $x_0$-parameterization-predicting the mean of distribution of $x\_0$ given $x\_t$ using $\mu_\theta$ (i.e. $p\_\theta(x_0 | x_t)$), and then use conditional posterior distribution $q({x}\_{t-1} \mid {x}\_t, {x}\_0)$ to sample the next step . You can verify that it satisfies the formula written in D3PM $p\_\theta(x\_{t-1} | x\_t) \propto \sum\_{\hat{x}} q(x\_{t-1} | x\_t, \hat{x}) p\_\theta(\hat{x} | x\_t)$.
> > >
> > > Importantly, this is **a defnition of $x_0$-parameterization instead of a direct factorization**, as explained in Section 3.3 of D3PM, where the authors justify this as a practical approximation for learning the reverse process.
> > >
> > > Many subsequent works, including MDLM [8] and MD4 [9], adopt the same notation style as ours, as **it is more concise and interpretable**. We also encourage reviewers to examine relevant codebases (e.g., MDLM’s), which aggregate prior implementations including D3PM. These can offer a more concrete understanding of how the D3PM sampling procedure corresponds exactly to our written formulation.
> > >
> > > We notice that reviewer have confusing the usgae of $p$ and $q$ in the review, and making the formula written like $p(x_{t-1} | x_t) = \sum_{\hat{x}} p(x_{t-1} | x_t, \hat{x}) p_\theta(\hat{x} | x_t)$, looking like a decomposition. **We kindly remind reviewer that in variational inference framework like discrete diffusion process, $p$ and $q$ cannot be used interchangeably** and they actually refer to different concepts. To clarify: $p_\theta\left(x_{t-1} \mid x_t\right)$ is the learned reverse process, which can actually adopt arbitrary form in principle (but $x_0$-paramerization is used in practice), and $q\left(x_{t-1} \mid x_t, x_0\right)$ is the true posterior from the forward noising process. We suspect this is causing your misunderstanding.
> > >
> > > Just let me know if you have any problems in understanding their equivalanece—we’d be happy to elaborate.
> > >
> > >
> > >
> > > [8] Sahoo S, Arriola M, Schiff Y, et al. Simple and effective masked diffusion language models[J]. Advances in Neural Information Processing Systems, 2024, 37: 130136-130184.
> > >
> > > [9] Shi J, Han K, Wang Z, et al. Simplified and generalized masked diffusion for discrete data[J]. Advances in neural information processing systems, 2024, 37: 103131-103167.

---

> > > > ### Comment · Reviewer_WEg3 · 2025-08-05
> > > >
> > > > Apologies for the inconsistency in notation in my previous comment. Replacing $p$ with $q$, one recovers Eq. 4 from D3PM:
> > > >
> > > > $p_\theta(x_{t-1} | x_t) = \sum_{\hat{x}} q(x_{t-1} | x_t, \hat{x}) p_\theta(\hat{x} | x_t)$
> > > >
> > > > Eq. 4 is not a definition in the sense that it follows directly from the law of total probability. Note that D3PM uses the symbol $\propto$ in Eq. 4, but the equality actually holds. This is clarified in the final equation of the same paragraph, which matches Eq. 4 with $k=1$.
> > > >
> > > > That said, I believe your presentation would benefit from explicitly showing how to derive Eq. 3 (or Eq. 6) starting from this expression. Making this connection more transparent would improve clarity and help avoid confusion.

---

> > > ### Author Response · Authors · 2025-08-03
> > >
> > > **3a. The core contribution of the method is the conditioning on the full denoising sequence, rather than only the last denoised step. A proper ablation is needed to isolate this effect. Table 5 only reports results for window sizes 3, 5, and 8. It should include at least size 1 (2 would also be informative). Moreover, the results do not clearly show that increasing window size improves performance. The gains appear marginal and are reported without error margins.**
> > >
> > > First, we would like to note that **Table 5 reports perplexity (PPL)** and **bits-per-dimension (BPD)**—both of which are **likelihood-based metrics** that evaluate how well the model fits the test data. These are not sampling-based metrics such as generative perplexity (GenPPL), and therefore they **do not vary across different runs**. As such, reporting error bars is not meaningful or standard practice in this setting. This convention is consistent with prior work, including D3PM, MDLM, MD4, and UDLM.
> > >
> > > Regarding the impact of the context window size, we respectfully disagree that the improvements are marginal. For example, as the context window size increases from 3 to 8, **BPD improves from 1.791 to 1.740**, consistently across settings. While the absolute numbers may seem small, we emphasize that **perplexity and BPD are sensitive metrics**—even improvements on the order of 0.05 on datasets like text8 are considered significant. For reference, the performance gain from baseline D3PM to more advanced models like SEDD or MDLM is also around 0.05 on this benchmark.
> > >
> > > As for the additional context window sizes (e.g., size 1 or 2), we agree these would be valuable additions. However, due to the **tight timeline of the rebuttal phase** (with only three days remaining) and the **official guidelines during the reviewer-author discussion**, we regret that we are unable to include new results at this stage. We appreciate your understanding on this matter.
> > >
> > > That said, we hope the following results from our initial rebuttal (point 3) can already help to address your concern regarding the benefit of conditioning on the full denoising trajectory:
> > >
> > > | Model | Reverse Process Condition | GenPPL on LM1B↓ |
> > > | -| -| -|
> > > | SID[1]|$x_t$-only|185.32|
> > > | CaDDi (Our Model)|trajectory| 142.51|
> > >
> > > Here, SID can be interpreted as a counterpart of CaDDi where the reverse process conditions only on $x_t$. The results clearly show that conditioning on the trajectory leads to significantly improved generation performance, as measured by generative perplexity.
> > >
> > > We appreciate your suggestion and will incorporate a discussion of this limitation in the revised version. Thank you again for your thoughtful feedback.

---

> > > ### Author Response · Authors · 2025-08-03
> > >
> > > **3b. The ability to revisit past states is not unique to full-sequence conditioning. Non-Markovian forward processes (e.g., [1]) can also enable this. Further justification is needed.**
> > >
> > > We acknowledge that non-Markovian forward processes, such as in [1], can indeed **remask** tokens from earlier timesteps. However, that is not the notion of "**revisiting**" we are referring to. Specifically, our claim concerns the **reverse process**: under a non-Markovian forward process, incorporating the **denoising trajectory** into the reverse process—i.e., modeling $p_\theta(x_{t-1} \mid x_{t:T})$ instead of just $p_\theta(x_{t-1} \mid x_{t})$ yields a strict mutual information gain $I(x_{t-1};x_{t+1:T}|x_t) > 0$. This implies that the existing denoising steps contain useful signals for inferring earlier states.
> > >
> > > Because our model explicitly conditions on this denoising trajectory, the reverse process can "**revisit**" and utilize information from subsequent states when predicting $x_{t-1}$. Intuitively, this enables the transformer’s attention mechanism to automatically identify and leverage informative parts of the trajectory—even from states that would otherwise be inaccessible under a Markovian assumption.
> > >
> > > This revisiting mechanism is thus not merely a byproduct of the forward remasking schedule, but a learnable and explicit component of our reverse process design. It aligns well with the inductive biases of transformer architectures, which excel at modeling long-range dependencies.
> > >
> > > We will clarify this distinction in the revised manuscript and update our language to more precisely reflect this form of revisiting. Thank you again for the opportunity to sharpen this explanation.

---

> > > ### Author Response · Authors · 2025-08-03
> > >
> > > **7 (New): My intuition is that conditioning on the full denoising sequence concentrates probability mass around distribution modes—akin to reducing the temperature parameter. This may not improve distribution modeling, but rather sharpen it. Could the authors comment?**
> > >
> > > Thank you for the thoughtful observation—this is indeed an interesting perspective. However, we respectfully disagree with the interpretation that conditioning on the full denoising sequence merely sharpens the distribution akin to reducing the temperature.
> > >
> > > First, as shown in our **Text8 perplexity results (Table 2)**, this benchmark evaluates the model's **likelihood-based performance**—i.e., how well the model's predicted distribution matches the test set distribution. It **does not involve sampling** (unlike generative perplexity or GenPPL) and is therefore **unaffected by sampling temperature**. In fact, artificially sharpening the distribution (e.g., by lowering entropy) typically **worsens** such metrics, as it increases the divergence from the ground truth distribution. Thus, improvements in perplexity and bits-per-dimension reflect **better distribution modeling**, not just concentration around modes.
> > >
> > > Second, in our **generation-focused evaluations on LM1B (Table 1)**, we report both **quality and diversity** metrics, including **entropy**. The entropy values for our method are comparable to those of baselines, suggesting that our model does not simply collapse onto dominant modes or lose diversity.
> > >
> > > Given this, we believe the observed improvements stem not from reducing entropy or over-sharpening, but from the model’s ability to leveraging richer contextual information by revisiting past states across the denoising trajectory. This leads to both improved quality and faithful distribution modeling.
> > >
> > > We will clarify this distinction in the revised version of the paper. Thank you again for raising these important points!

---

> ### Comment · Reviewer_WEg3 · 2025-08-05
>
> 3b. Thank you for the clarification

---

> ### Author Response · Authors · 2025-08-06
>
> No need to apologize! As authors, we are alway pleasant to clarify and help reviewers understand in rebuttal stage. As for the issue, **we respectively maintain our position that  that Eq. 4 in D3PM represents a definition of x0-paramerization instead of a derivation**. We reached out (anonymously) to the authors of D3PM, who confirmed the following about D3PM:
>
> "*...I would say that equation 4 is a definition rather than a derivation. there are other forms. And we pick this posterior integrated form because it turns out to work better. DDPM also does someting similar in continious space...*",
>
> This aligns with our original explanation. We notice in your latest reply you state that $p_{\theta}(x_{t-1}|x_t) = \sum_{\hat{x}}q(x_{t-1}|x_t, \hat{x})p_{\theta}(\hat{x}|x_t)$ holds, which lead you to a conclusion that it's a derivation. **But this is generally not true. The equation only hold when $p_{\theta}=q$**. This equation is a definition instead of a direct derivation. We doubt it is still this part causing your confusion.

---

> ### Author Response · Authors · 2025-08-06
>
> We also want to kindly remind you the motivaion of $x_0$-parametrization (both original D3PM and our non-Markovian discrete diffusion process). If we go back to the original definition of latent variable models. Suppose we have a fixed forward process $q$ resulting in a series of latent variables $x_{1:T}$, the ELBO can then be written in form of
>
> $$
> \mathcal{L}\_{\theta}^{\text{ELBO}} = \mathbb{E}\_{{x}\_{1:T} \sim q({x}\_{1:T}|{x}\_0)}\left[\sum\_{t=1}^{T}\log p\_{\theta}({x}\_{t-1}|{x}\_{t:T}) + \log p\_{\theta}({x}\_T) - \log q({x}\_{1:T}|{x}\_0)\right],
> $$
>
>
> This can be rewritten as:
>
> $$
> \mathcal{L}\_{\theta}^{\text{ELBO}} = \mathbb{E}\_{{x}\_{1:T} \sim q({x}\_{1:T}|{x}\_0)}\left[\sum\_{t=2}^{T}\log \frac{p\_{\theta}({x}\_{t-1}|{x}\_{t:T})}{q(x\_{t-1} | x\_{t:T}, x\_0)} + \log \frac{p_{\theta}({x}\_T)}{q(x\_T | x_0)} + \log p\_\theta({x}\_0|{x}\_{1:T})\right],
> $$
>
> **Note that the above equation holds generally and is not under any specfic assumption. $p_\theta$ here is a learned reverse process. It can actually adopt any form or even directly use a neural network to output the value**
>
> 1. Now we consider the **Marokvian discrete diffusion case**, which means that $q(x_{t} | x_{0:t-1}) = q(x_t | x_{t-1})$. Under this assumption, one can show that $q(x_{t-1} | x_{t:T}, x_0) = q(x_{t-1} | x_t, x_0)$ ($x_{t-1}$ and $x_{t+1:T}$ are conditionally independent when given $x_t$ and $x_0$). Then the above ELBO can be further simplified as
>
> $$
>  \mathcal{L}\_{\theta}^{\text{ELBO}} = \mathbb{E}\_{{x}\_{1:T} \sim q({x}\_{1:T}|{x}\_0)}\left[\sum\_{t=2}^{T}\log \frac{p\_{\theta}({x}\_{t-1}|{x}\_{t:T})}{q(x\_{t-1} | x\_t, x\_0)} + \log \frac{p\_{\theta}({x}\_T)}{q(x\_T | x\_0)} + \log p\_\theta({x}\_0|{x}\_{1:T})\right],
> $$
>
> In that case, **a natural definition of $p\_{\theta}({x}\_{t-1}|{x}\_{t:T})$ is to make it Markovian and align with $q$, resulting $p\_{\theta}({x}\_{t-1}|{x}\_{t:T}) := p\_{\theta}({x}\_{t-1}|{x}\_{t}) := q({x}\_{t-1}|{x}\_{t}, x_0 = \mu\_\theta(x\_t, t))$**. So we only need to train a denoiser $\mu\_\theta(x\_t, t)$ to predict $x\_0$. This is the hidden motivation of $x\_0$-parameterization of D3PM. **Note that this is a definition instead of a derivation, one can still use other forms (e.g. directly predicting $x\_{t-1}$)**. It's just empirical results indicating it's better.
>
>
> 2. Now we consider our **non-Markovian discrete diffusion case**: In our case, the forward kernel is designed to be non-Markovian, so $q(x_{t} | x_{0:t-1}) = q(x_t | x_{t-1})$ does not hold anymore. Instead, since we use an independent noising kernel across timesteps, we have $q(x_{t-1} | x_{t:T}, x_0) = q(x_{t-1} | x_0)$. In this case, we have ELBO as
>
>
> $$
>  \mathcal{L}\_{\theta}^{\text{ELBO}} = \mathbb{E}\_{{x}\_{1:T} \sim q({x}\_{1:T}|{x}\_0)}\left[\sum\_{t=2}^{T}\log \frac{p\_{\theta}({x}\_{t-1}|{x}\_{t:T})}{q(x\_{t-1} |  x\_0)} + \log \frac{p\_{\theta}({x}\_T)}{q(x\_T | x\_0)} + \log p\_\theta({x}\_0|{x}\_{1:T})\right],
> $$
>
> So correspondingly, a natural extension is to **define** $p\_\theta$ as $p\_{\theta}({x}\_{t-1}|{x}\_{t:T}):=q(x\_{t-1} | x\_0=\mu\_\theta(x\_{t:T}, t))$. Here **the reverse process $p$ is kept non-Markovian (containing history $x_{t+1:T}$)**, as in case of non-Markovian forward kernel, $x\_{t-1}$ and $x\_{t+1:T}$ are **not** conditionally independent when given $x_t$ and $x_0$. This suggests that **trajectory $x_{t+1:T}$ also contain information about $x\_{t-1}$**
>
> So generally, imagine $p_\theta$ as a learned reverse process we want to maximize ELBO. There is never a derivation to get its strict form. It's just intuitively and empirically $x_0$-parameterization works better.
>
> We hope the above has helped your understanding why $x_0$-parameterization is a practical design for the definition of $p\_\theta$ in D3PM and our Non-Markovian discrete diffusion. If you still have confusion in understanding anything related to variational inference, ELBO in discrete diffusion, $x_0$-parametrization, we are happy to elaborate.
>
> If reviewers still have doubts about the formulation of D3PM, also feel free to reach out to any author of these established prior work (e.g. D3PM, MDLM, MD4) anonymously to verify all of the above statements. From our experience, they are open and friendly about discussing these technical details.
>
> Thank you once again for your feedback!

---

> > ### Comment · Reviewer_WEg3 · 2025-08-06
> >
> > Thanks a lot for these clarifications, and sorry for the long discussion. You are right. As it is the core of your method, I think it would help to push these explanations in the paper (in the appendix if space is missing).
> >
> > You have now answer all my concerns and I will increase my score in consequence.

---

### Official Review · Reviewer_cgB8 · 2025-07-01

**Clarity:** 3
**Significance:** 3
**Originality:** 3
**Rating:** 5
**Confidence:** 3

**Summary:**

This paper introduces a new discrete diffusion framework, CaDDi, with its variant CaDDi-AR. This framework removes the Markovian assumption and regards the noise as independently injected into the clean data. From the new assumption, the author derives an inference process and a specific training objective through the ELBO. Built upon the mathematical formulation and the observation of the connection between non-Markovian diffusion and the causal language model, the author proposes to integrate token order and timesteps into a single transformer model. To account for both dimensions, the authors also propose a 2D Rotary Positional Encoding method. The CaDDi-AR factorizes over the token space to capture the true posterior distribution and uses speculative decoding to reduce the computation. Since the causal language model can be regarded as a special case of the CaDDi-AR, pre-trained LLMs can be fine-tuned to serve as the backbone of the CaDDi-AR. Comprehensive experiments have been done to prove its effectiveness. The experiments include 5 baselines, on 3 open-source LLMs, on a billion-word dataset, Text8 dataset, and a bunch of reasoning benchmarks. The overall experimental results suggest that CaDDi is an effective discrete diffusion framework across many models and datasets.

**Questions:**

N/A.

**Ethical Concerns:**

["NO or VERY MINOR ethics concerns only"]

**Final Justification:**

The rebuttal has addressed most of my concerns. The paper is a solid contribution and is suitable for the NeurIPS.

**Limitations:**

N/A.

**Paper Formatting Concerns:**

N/A.

**Quality:**

3

**Strengths And Weaknesses:**

Strength:
1. The core idea behind CaDDi is great. It's a non-Markovian approach to discrete diffusion, meaning it conditions the denoising on the entire generation history, not just the last step. This is a well-motivated way to tackle the error accumulation problem that plagues standard Markovian models. The theory holds up, and the training objective is straightforward.
2. The 2D RoPE is a really clever architectural choice. It neatly handles both token position and the diffusion timestep within a regular decoder-only Transformer. The best part is that this makes it backward-compatible, so you can fine-tune existing LLMs without having to change their architecture.
3. The authors clearly anticipated the potential for high computational cost with the token-level version (CaDDi-AR). Their "semi-speculative decoding" strategy is a smart fix. By using the prediction from the previous timestep as a draft for the current one, it cuts down on sampling steps and makes the whole process more efficient.
4. The experiments are solid. CaDDi was benchmarked against a bunch of SOTA discrete diffusion models (D3PM, SEDD, MDLM, DFM) and consistently came out on top in both perplexity and reasoning tasks.

Weakness:
1. My main issue is with the comparison to standard autoregressive models. While CaDDi is clearly better than other diffusion models, it's may inherently slower than a single-pass AR model because it requires T sampling steps. The paper brings up semi-speculative decoding to help with this, but there's no direct, wall-clock time comparison to a similarly-sized AR model. I'm left wondering how practical CaDDi-AR really is. How close does it get to the speed of a fine-tuned AR baseline for the same quality? This is a key question for real-world use.
2. The non-Markovian design means conditioning on the history of latent states. This could create a massive input sequence for the transformer, especially with a large T. The paper doesn't really dig into how well the model scales with longer context lengths, which seems like an important limitation to explore.

---

> ### Author Rebuttal · Authors · 2025-07-31
>
> We thank the reviewer for their constructive feedback. We appreciate that they recognize the core idea of CaDDi, and we are grateful for their positive remarks on the architectural design, the semi-speculative decoding strategy, and the experiments.
>
> 1. **Time efficiency of CaDDi-AR**
>
> We thank the reviewer for highlighting the issue of CaDDi’s time efficiency. We agree that, due to the iterative refinement nature of the diffusion process, multiple passes are unavoidable—this being a key distinction from autoregressive (AR) models. Similarly, CaDDi-AR, which interpolates between the diffusion and AR paradigms, also inherits this increased time complexity. To address this, we proposed a semi-speculative decoding strategy, which significantly accelerates sampling and mitigates the computational overhead.
>
> We would like to highlight that CaDDi-AR is a special case within the broader CaDDi framework, which is designed as a flexible interpolation between discrete diffusion and autoregressive (AR) modeling. CaDDi can be tuned to lean more toward the diffusion paradigm or the AR paradigm, depending on the application needs. This flexibility reflects an inherent trade-off between sampling efficiency and modeling expressiveness, allowing practitioners to balance performance and speed according to their specific use case.
>
> Finally, to demonstrate the time complexity of CaDDi, we include below the Number of Function Evaluations (NFE) comparison between different versions of CaDDi and MDLM with varying diffusion steps, together with their performance on the LM1B dataset. We see that while CaDDi-AR uses a high number of NFEs, semi-speculative decoding strategy reduces the numer to a reasonable regime without a significant drop in performance. Also, CaDDi with 64 steps already achieves better performance than MDLM with 1000 NFE (and with larger FLOPs). Therefore, practitioners are safe to choose the specific CaDDi model they could afford based on their need and resources.
>
> | Model                                     | NFE  | GenPPL on LM1B↓ |
> | ----------------------------------------- | ---- | --------------- |
> | MDLM-64                                   | 64   | 199.45          |
> | MDLM-500                                  | 500  | 188.93          |
> | MDLM-1000                                 | 1000 | 179.41          |
> | CaDDi                                     | 64   | 142.51          |
> | CaDDi-AR                                  | 8192 | 137.82          |
> | CaDDi-AR (with semi-sepculative decoding) | ~656 | 139.80          |
>
>
> 2. **Context length**
>
> We appreciate the reviewer for highlighting a key distinction of our method: conditioning on the full history trajectory rather than a separate time step. Indeed, without special treatment, the context length would grow with the timestep t, potentially making it computationally expensive or unstable for large T. In Appendix C.1 ("Context Window and Latent Compression"), we discuss this practical limitation and introduce latent truncation and trajectory re-composition as strategies to compress the latent trajectory to fit within a bounded context window. Specifically, latent truncation (Section C.1.1) limits the conditioning to the most recent m timesteps, which directly controls the computational cost. We have performed an ablation study on the context window size (Section G.2, Table 5 and Figure 9), showing that a moderate window size offers a favorable balance between performance and computational cost.

---

> > ### Comment · Reviewer_cgB8 · 2025-08-05
> >
> > Thank you for your responses, which have addressed most of my concerns. My primary remaining question is about computational efficiency. Since both the number of sampling steps and the context length increase the computational cost, I would like to see a FLOPs comparison in your revision. This analysis should demonstrate that the increased computational demand worth it, when compared to both AR  and diffusion models. I will maintain my current score and recommendation for acceptance.

---

> > > ### Author Response · Authors · 2025-08-06
> > >
> > > We thank the reviewer for their constructive feedback and continued engagement. We agree that including a FLOPs comparison in addition to the current NFE analysis would provide a more comprehensive view of computational efficiency. We will incorporate this into the revised version of the paper. Thank you again for your thoughtful suggestions and support.

---

### Official Review · Reviewer_wro5 · 2025-07-05

**Clarity:** 2
**Significance:** 2
**Originality:** 2
**Rating:** 4
**Confidence:** 3

**Summary:**

This paper introduces a Non-Markovian discrete diffusion framework for text generation. Unlike conventional discrete diffusion models that rely on the Markovian assumption, CaDDi conditions each denoising step on the full generative trajectory.

Furthermore, CaDDi is introduced as a causal version of discrete diffusion, using transformer architecture as a natural bridge. It unifies sequential and temporal reasoning.

Experimental results demonstrate superior generative quality and robustness on several benchmarks (LM1B, Text8, Amazon Polarity, reasoning tasks) compared to existing discrete diffusion baselines, while retaining compatibility with pretrained LLMs.

**Questions:**

In addition to weaknesses,

1. Could the authors further quantify the computational cost of CaDDi and CaDDi-AR compared to traditional AR models and Markovian discrete diffusion in both training and inference?

2. What is the sequence length in the experiments? Have the authors tested CaDDi on longer sequences, and how does it scale in terms of memory, computation, and quality?

**Ethical Concerns:**

["NO or VERY MINOR ethics concerns only"]

**Final Justification:**

The reviewer have addressed all my concerns in their rebuttal. I adjusted my score due to that the discussion with prior work is sufficient.

**Limitations:**

I do not see the discussion on limitations in this paper.

**Quality:**

2

**Strengths And Weaknesses:**

Strength:

- The paper introduces a 2D positional encoding, which is novel.
- The paper introduces a semi-speculative decoding to speed up inference.
- Ability to reuse pretrained LLMs for diffusion tasks broadens the applicability of diffusion-based approaches.
- Demonstrates clear improvements in generative perplexity.

Weaknesses:

- The authors did not point out that the idea of Non-Markovian discrete diffusion is not novel. For example, [1] first propsed a Non-Markovian discrete diffusion, in addition to allowing $p(x_t|x_{t-1}, ... x_{0})$ to be conditioned on the whole sequence, it is also aware of the sequencial and temporal relationship of Non-Markovian discrete diffusion (using transition time $\tau$ to gain faster inference).
- Due to the above reason, there is a lack of proper reference to prior work on Non-Markovian discrete diffusion models like DNDM, as well as Non-Markovian diffusion models like DDIM.
- Despite semi-speculative decoding, it seems that inference remains slower than discrete diffusion models, since the complexity is L*T. The authors did not discuss the remaining computation overhead.
- The "VERIFY" and "CORRECT" process in Algorithm 2 has never been defined in this paper.
- Algorithm 2 is never referenced. It seems like a typo (Line 216 says "see Algorithm 3")

[1] Chen, Zixiang, et al. "Fast Sampling via Discrete Non-Markov Diffusion Models with Predetermined Transition Time." NeurIPS 2024.

---

> ### Author Rebuttal · Authors · 2025-07-31
>
> We sincerely thank the reviewer for their thorough and insightful feedback on our paper. The comments are highly valuable and will help us significantly improve the clarity, completeness, and impact of our work. We are encouraged that the reviewer found our 2D positional encoding and semi-speculative decoding to be novel and appreciated the performance improvements and the ability to reuse pretrained LLMs.
>
> Below, we address each of the reviewer's points and detail our planned revisions.
>
> **1. & 2. Novelty and Referencing of Non-Markovian Discrete Diffusion DNDM**
> Thank you for highlighting the need for a more prominent comparison with DNDM. Actually we did include a discussion of related work including DNDM in our 22-page appendix (in supplementary material), but we understand that supplementary material can be easily missed and agree that this comparison is crucial for the main paper. We appreciate the opportunity to clarify the distinction and re-emphasize our specific contributions.
>
> - **Regarding DNDM**: While both CaDDi and DNDM are non-Markovian, they address different goals with distinct methodologies. DNDM's primary contribution is achieving **fast, training-free sampling** by derandomizing the generation process with a predetermined transition time. In contrast, CaDDi leverages the non-Markovian structure to **improve generative quality and robustness** by explicitly conditioning the denoiser on the generative trajectory.
> Therefore, the conceptual overlap is limited to the relaxation of the Markovian assumption. Our core contributions are orthogonal to theirs and focus on a novel modeling paradigm:
>     1.  A unified causal framework that models both sequential and temporal dimensions in a single transformer.
>     2. A novel 2D Rotary Positional Encoding (RoPE) designed for this unified structure.
>     3. A direct method for adapting pretrained LLMs to the discrete diffusion paradigm.
>
> Thanks again for your suggestions. We will move the discussion of DNDM from the appendix to the main body of the paper, expanding the explaination.
> - **Regarding DDIM**: DDIM is a foundational work for non-Markovian sampling in **continuous** diffusion models. Our paper is focused on the discrete domain, where the challenges and solutions are fundamentally different (e.g., operating on categorical variables, different kernel designs, and no direct equivalent of an ODE formulation).
>
> **3. Inference Speed and Computational Overhead (Complexity L×T) & Quantifying Computational Cost (Training and Inference):**
> We acknowledge the reviewer's point regarding the inference complexity and the need for a more detailed quantification of computational cost. While the theoretical complexity for naive CaDDi-AR's token-level autoregressive generation within each diffusion step is indeed L×T, our semi-speculative decoding strategy is designed to significantly reduce this in practice. The exact degree of reduction cannot be theoretically derived as it's an algorithm that dynamically skips steps; however, it consistently results in a substantially lower effective number of function evaluations (NFE) than L×T. To further quantify the inference computational cost, we provide the following table on the LM1B dataset during generation:
> | Model | NFE | GenPPL on LM1B↓ |
> | -| -| -|
> |MDLM-64|64|199.45|
> |MDLM-500|500|188.93|
> |MDLM-1000|1000|179.41|
> |CaDDi|64|142.51|
> |CaDDi-AR|8192|137.82|
> |CaDDi-AR (with semi-sepculative decoding)|~656|139.80|
>
> As shown, semi-speculative decoding significantly reduces the NFE for CaDDi-AR while maintaining competitive performance, highlighting its effectiveness as a speedup technique. It is also worth noting that the non-AR version of CaDDi (with complexity O(L)) with 64 steps already achieves better performance than MDLM with 1000 NFE (and with larger FLOPs). CaDDi-AR is a strategy to further enhance performance, which inherently entails a higher computational cost, and semi-speculative decoding is specifically designed to alleviate this additional overhead.
>
> Regarding the overall overhead of computation in training, we discuss this practical limitation in Appendix Section C.1 ("Context Window and Latent Compression") and introduce latent truncation and trajectory re-composition as strategies to compress the latent trajectory to fit within a bounded context window. Specifically, latent truncation (Section C.1.1) limits the conditioning to the most recent m timesteps, which directly controls the computational cost in both training and inference. We have already performed an ablation study on the context window size (Section G.2, Table 5, Figure 9), showing that a moderate window size offers a favorable balance between performance and computational cost. These strategies ensure that CaDDi achieves better performance while not excessively sacrificing computational cost.
>
> **4. Terminology Clarification of "VERIFY" and "CORRECT" in Algorithm 2:**
> "VERIFY" and "CORRECT" refer to the standard operations within speculative decoding, adapted for CADDi-AR. Specifically:
> - **VERIFY**($p_0$, $\mathbf{x}\_{1:T}$, ${\hat{x}}\_0^{\text{prev}}$): This step takes the model $p_0$, the current latent trajectory $\mathbf{x}\_{1:T}$, and the draft sequence $\hat{x}_0^\text{prev}$ (which is the predicted clean sequence from the previous timestep). It evaluates the probability of each token in ${\hat{x}}_0^\text{prev}$ given the context and determines the longest prefix of ${\hat{x}}_0^\text{prev}$ for which the tokens are accepted based on a specified confidence threshold (In this work, we use a criterion similar to Nucleus Sampling, rejecting samples based on the probability of choosing the same token between this timestep and the prior timestep. If the probability is higher, no rejection happens). The output $\tilde{i}$ is the index of the first token that is rejected or where the draft ends.
> - **CORRECT**($p_0$, $\mathbf{x}\_{1:T}$, ${\hat{x}}\_0^{\text{prev}}$): This step takes the same inputs. It returns the accepted prefix of $\hat{x}\_0^\text{prev}$ (i.e., ${\hat{x}}\_0^{0:\tilde{i}-1}$) and prepares the model to resume autoregressive sampling from the position $\tilde{i}$. This process is analogous to how speculative decoding "corrects" a proposed draft by either accepting it fully or partially and then continuing generation from the point of divergence. We will ensure these operations are precisely described in Section 4.2 to enhance clarity.
>
> **5. Typo in Algorithm Reference (Line 216):**
> Thanks for pointing this out! We will fix it in the next revision.
>
> **Q1.Further Quantification about the Computational Cost of CaDDi and CaDDi-AR**
> See Point 3.
>
> **Q2. What's the Sequence Length in Experiment and How Does it Scale on Longer Sequences?**
> The sequence length used for the LM1B dataset is 128 tokens, and for the Text8 dataset, it is 256 tokens. These dataset configurations are standard, consistent with prior discrete diffusion work such as D3PM and MDLM.
>
> For CaDDi specifically, the primary factor affecting memory and computation for longer sequences is the transformer's self-attention mechanism, which scales quadratically with sequence length. To address this, Appendix C (Model Implementation Details) discusses "Latent Truncation" and "Trajectory Re-composition" as strategies to manage the context window and handle longer generative trajectories. These techniques effectively compress $x_{t:T}$ to fit within the model's context. These strategies allow CaDDi to scale to longer sequence.
>
> It is also important to note that most existing standard discrete diffusion models focus on limited sequence length generation due to their encoder structures. However, recent work, such as Block Diffusion [1], has been proposed to extend beyond these limitations. Integrating such ongoing efforts with our framework to further scale to longer sequences presents an interesting future direction.
>
>
> [1]. Arriola M, Gokaslan A, Chiu J T, et al. Block diffusion: Interpolating between autoregressive and diffusion language models[J]. arXiv preprint arXiv:2503.09573, 2025.
>
> We thank the reviewer once again for their valuable feedback! We believe these revisions will significantly strengthen the paper by addressing the reviewer's concerns comprehensively and transparently.

---

> > ### Author Response · Authors · 2025-08-07
> >
> > Thank you very much for the time and thoughtful attention you have devoted to reviewing our manuscript. We hope our responses have satisfactorily addressed remaining concerns. With fewer than two days left in the review period, please don’t hesitate to let us know if any further clarification is needed—we would be happy to respond promptly.

---

> > ### Comment · Reviewer_wro5 · 2025-08-09
> >
> > Dear Authors,
> >
> > Thank you for your detailed response. As my main concern has been addressed, I maintain my view that the techniques for handling encoding and decoding are novel, and I will accordingly raise my score.
> >
> > Reviewer wro5

---

### Official Review · Reviewer_qCqR · 2025-07-07

**Clarity:** 3
**Significance:** 4
**Originality:** 3
**Rating:** 5
**Confidence:** 3

**Summary:**

This paper proposes a novel diffusion model, called CaDDi, for discrete sequences. Instead of relying only on the previous diffusion step, it leverages all diffusion history to provide a more stable discrete diffusion model. Notably, when the diffusion step T=1 it can be regarded as an ordinary causal language model, thus allowing finetuning existing large language model using the power of diffusion model.
Extensive experimental results indicate that the proposed CaDDi has a better performance than previous discrete diffusion models. While it is slightly behind ordinary causal LLMs without finetuning, this is an important step towards useful diffusion model for discrete sequences.

**Questions:**

Nothing.

**Ethical Concerns:**

["NO or VERY MINOR ethics concerns only"]

**Limitations:**

Nothing.

**Paper Formatting Concerns:**

Nothing.

**Quality:**

4

**Strengths And Weaknesses:**

See above. I found no clear weaknesses in this approach.
While pages are limited, I would like to see some practical texts where the proposed CaDDi works better than ordinary causal LMs or other discrete diffusion models.

---

> ### Author Rebuttal · Authors · 2025-07-31
>
> We sincerely thank Reviewer qCqR for their thoughtful and encouraging review. We are glad that the reviewer recognizes the novelty, significance, and technical quality of our work. We appreciate the comment regarding practical use cases where CaDDi outperforms standard causal LMs or existing discrete diffusion models. While such examples are constrained by space in the current submission, we agree this is an important direction and will add some additional illustrative use cases in the manuscript.

---

### Official Review · Reviewer_CSEv · 2025-07-09

**Clarity:** 2
**Significance:** 2
**Originality:** 2
**Rating:** 2
**Confidence:** 4

**Summary:**

The authors argue that the Markovian assumption in discrete diffusion models is a key limitation, as it can lead to error accumulation during sampling. To address this, they propose a novel algorithm called CaDDi, which conditions the denoising model’s predictions on the entire generation trajectory. Specifically, they modify the transformer architecture to incorporate the full diffusion trajectory as input. Empirically, CaDDi achieves competitive performance on natural language benchmarks.

**Questions:**

See weaknesses above.

**Ethical Concerns:**

["NO or VERY MINOR ethics concerns only"]

**Final Justification:**

The ancestral sampler in masked diffusion models cannot update its predictions. This paper proposes a method to allow the diffusion model to update its predictions. However, **it does not compare against existing predictor–corrector methods [1, 2, 3]**, which achieve the same goal without changing the model architecture or requiring any fine-tuning.


For this reason, I strongly believe the paper should be rejected.

I would like to maintain my score for the following reasons:
1. Computational cost: CaDDi achieves a better perplexity than MDLM by using 5× more FLOPs (ignoring the quadratic attention cost in sequence length for now). A fair comparison would be an MDLM with a similar FLOP budget—equivalent to ~5× more parameters, since FLOPs ≈ 6 × (number of parameters) × (training tokens). **This method will not scale to longer context lengths, as attention will become the limiting factor**.

2. Fine-tuning requirement:
> By unifying temporal and sequential (spatial) dependencies within a single causal transformer architecture, CaDDi enables direct reuse of pretrained LLM weights without requiring architectural changes. To our knowledge, this capability has not been explored in prior work on non-Markovian discrete diffusion.

While this is presented as a strength, CaDDi still requires fine-tuning an existing LLM. By contrast, **predictor–corrector approaches [1, 2, 3] require no fine-tuning** at all.


3.	Architectural changes: Other reviewers seem particularly impressed by the 2D RoPE embeddings introduced in this paper. **However, this is actually a disadvantage**: predictor–corrector approaches achieve the same functionality without any architectural modifications.


4. Weak experiments: LM1B with a GPT-2 tokenizer yields ~30 unique tokens per sentence, each contextually distinct, making generative perplexity meaningless. Overall, the paper relies on toy datasets and lacks comparison to sensible baselines.



### References:
[1] Wang et al., 2025 "Remasking Discrete Diffusion Models with Inference-Time Scaling"

[2] Campbell et al. 2022, "A Continuous Time Framework for Discrete Denoising Models"

[3] Gat et al., 2025, "Discrete Flow Matching"

**Limitations:**

see weaknesses above.

**Quality:**

2

**Strengths And Weaknesses:**

### Strengths

The semi-speculative decoding sampler proposed in this paper is interesting.

### Weaknesses

1. In masked diffusion models, the state x_t inherently encodes the entire trajectory x_{t:T}, making the decoding sequence largely irrelevant. As shown in Table 2, CaDDi clearly lags behind MDLM in terms of perplexity, suggesting that the additional engineering to incorporate the entire trajectory does not provide a clear benefit.

2. It is unclear why the perplexity of the models on LM1B is not reported, given that the authors trained their model on this dataset. Perplexity remains the gold standard for comparing language models. Generative perplexity is typically used to evaluate samplers.

3. Table 1 is misleading:
(a) What does T refer to? Is it the sampling temperature or the number of time discretizations? What discretizations were used for the baselines such as MDLM, D3PM, and UDLM?
(a) What sampler was used for each method? If CaDDi uses the proposed semi-speculative decoding sampler, it would be helpful to report the sample quality from CaDDi using the ancestral sampler as well. This would allow the community to disentangle the improvements due to (1) the non-Markovian trajectory and (2) the semi-speculative decoding sampler, in comparison to MDLM.

4. Table 4 is missing major baselines. The authors should compare their method against MDLM, UDLM [2], and an autoregressive (AR) model trained with the same transformer architecture and a similar number of training steps. They can follow the steps in [2].



PS. This paper is riddled with typos. A major one is Algorithm 2 being incorrectly referenced Algorithm 3 in line 216.



References

[1] Sahoo et al., 2024, "Simple and Effective Masked Diffusion Language Models"

[2] Schiff et al., 2025, "Simple Guidance Mechanisms for Discrete Diffusion Models"

---

> ### Author Rebuttal · Authors · 2025-07-31
>
> We thank the reviewer for their thoughtful feedback and insightful comments. We appreciate the recognition of our semi-speculative decoding sampler as an interesting contribution. We address each of the raised weaknesses below.
>
> 1. **"In masked diffusion models, the state $x_t$ inherently encodes the entire trajectory $x_{t:T}$, making the decoding sequence largely irrelevant. As shown in Table 2, CaDDi clearly lags behind MDLM in terms of perplexity, suggesting that the additional engineering to incorporate the entire trajectory does not provide a clear benefit."**
>
> We respectfully disagree with the assertion that $x_t$ inherently encodes the entire trajectory $x_{t:T}$ in non-Markovian masked diffusion models. In CaDDi, noise is added independently at each timestep, conditioned only on the original data $x_0$. In this case, from the very nature of how trajectories are created, the state $x_t$ itself does NOT encode the entire trajectory before it.
>
> Now, we explain why the above formulation is necessary and helpful. As discussed in Section 3.1 and further elaborated in Appendix A.4 ("Failure to Remask Issue"), traditional Markovian discrete diffusion models, including MDLM, suffer from a "failure to remask" problem. Once a token is unmasked (i.e., $x_t \neq e_m$), it is deterministically copied to $x_{t-1}$ regardless of the model's prediction. This means the model cannot revise early mistakes, leading to error accumulation.
>
> Our non-Markovian approach ensures that **each state $x_t$ in the latent trajectory contains unique and complementary information**, rather than being a mere compression of the subsequent trajectory. This fundamental difference in the forward process is key to our non-Markovian formulation, which then allows the reverse process to condition on the historical trajectory $x_{t:T}$ (as described in Section 3.3 and Algorithm 1). This explicit conditioning enables the model to effectively revisit past states, thereby lifting the restrictive Markov constraint and leading to a more robust inference process that can self-correct errors.
>
> Regarding the perplexity comparison in Table 2, the reviewer states that CaDDi lags behind MDLM. We believe this observation may stem from an unfair comparison across different time discretizations, and we wish to clarify the following:
>
> - **Performance at Same Discretization (Table 2)**: We wish to emphasize that the reviewer might be wrongly comparing across different time discretizations. As noted in Appendix Section E.3, **likelihood estimation for diffusion models is sensitive to the number of discretization steps**. As clearly shown in Table 2 (rows shaded in gray), when comparing models at **the same 64-step discretization**, CaDDi achieves a BPD of $<1.41$ and Perplexity of $<2.66$, which is indeed **the best performance** among all discrete diffusion models evaluated at 64 steps (MDLM at 64 steps has BPD $\leq1.46$ and Perplexity $\leq2.75$). This demonstrates a clear benefit of our approach within comparable settings. Some baselines report results under continuous-time settings or with 1000-step discretizations, which are not directly comparable to our 64-step discrete evaluation. Since MDLM (and other continuous-time models) are trained in a continuous-time framework while our model operates in a discrete-time framework, there isn't a strict one-to-one correspondence for continuous frameworks. However, to further demonstrate our model's consistent superiority, here we trained an additional model with 256-step discretization (which is the same as the sequence length in the Text8 dataset). This model achieves a **BPD of ≤1.37** (better than MDLM even with $\infty$ time), further solidifying CaDDi's competitive performance.
> - **LM1B Results (Table 1)**: On the larger and more representative LM1B dataset, both CaDDi and CaDDi-AR consistently outperform MDLM in generative perplexity across all three language model oracles (GPT-2, Llama-2-7B, and Llama-3.2-3B) and various sampling temperatures. This demonstrates a clear benefit of our approach on a more challenging and widely used benchmark.
> - **Ablation Study on Robustness (Figure 4 and G.3)**: Our ablation study on inference robustness (Figure 4 and Figure 10 in Appendix G.3) clearly shows that CaDDi exhibits significantly stronger resilience to manually injected noise compared to MDLM and D3PM. This directly supports our argument that conditioning on the historical trajectory improves inference robustness and self-correction, which is a key benefit not captured solely by perplexity on Text8.
> Therefore, the overall empirical evidence, especially when fairly comparing at the same discretization on Text8, on LM1B, and in terms of robustness, strongly supports the benefits of our non-Markovian approach.
>
> **2. "It is unclear why the perplexity of the models on LM1B is not reported."**
>
> Thanks for raising this point. Our primary intention with the Text8 experiments was to demonstrate the likelihood modeling ability of our model, and we considered reporting additional perplexity metrics on LM1B to be somewhat redundant in that regard. Instead, we established the LM1B dataset as a generative task to evaluate the direct generative ability and quality of samples across different models.
>
> However, we appreciate your suggestion for including these results for completeness. We are pleased to provide the perplexity results for the LM1B dataset here
>
> |Model| PPL on LM1B↓|
> |-| -|
> |D3PM|≤77.50|
> |SEDD|≤32.79|
> |MDLM|≤23.00|
> |CaDDi|≤21.91|
>
> **3. Clarification Regarding Table 1**
>
> We sincerely acknowledge the reviewer for raising concerns. In the meantime, we highly recommend the reviewer to also refer to our comprehensive **22-page appendix in the supplementary material**. **This supplementary material has already covered most of the following concerns in greater detail**. But thanks to the suggestions, we recognize that some details might not have been sufficiently clear in the main text and will move relevant information from appendix to the main text in the next revision to improve clarity and accessibility.
>
> - **(a) What does T refer to? Is it the sampling temperature or the number of time discretizations?**: In Table 1, "T" refers to the sampling temperature (e.g., T=1, T=0.7, T=0.5). This is explicitly stated in the caption of Table 1. The number of time discretizations (diffusion steps) for CaDDi and CaDDi-AR is consistently 64 for the reported LM1B experiments, as stated in Appendix Section E.1 line 923. For baselines, Section E.1 notes: "*Note that models such as MDLM and SEDD are trained in continuous time, whereas our models operate in discrete time.*" For evaluation, "*All diffusion-based models use 64 denoising steps during inference.*"
> - **(b) What discretizations were used for the baselines such as MDLM, D3PM, and UDLM?** As mentioned above and in Appendix Section E.1, MDLM and SEDD are trained in continuous time. For evaluation, all diffusion-based models (including MDLM, D3PM, UDLM, SEDD, DFM) were evaluated using 64 denoising steps during inference for fair comparison on LM1B (Section E.1). For Text8 (Table 2), we explicitly state the steps/discretization for each model
> - **(c) What sampler was used for each method? If CaDDi uses the proposed semi-speculative decoding sampler, it would be helpful to report the sample quality from CaDDi using the ancestral sampler as well**. For CaDDi, we use the ancestral sampler as described in Algorithm 1. For CaDDi-AR, we employ the semi-speculative decoding sampler (Algorithm 2) to accelerate inference. We agree that disentangling the improvements is valuable. The performance reported for "CaDDi" in Table 1 uses the ancestral sampler (Algorithm 1), while "CaDDi-AR" uses the semi-speculative decoding sampler (Algorithm 2). The comparison between "CaDDi" and "CaDDi-AR" in Table 1 directly showcases the impact of the token-level autoregressive factorization and the semi-speculative decoding. CaDDi (block-level autoregressive) already outperforms MDLM, demonstrating the benefit of the non-Markovian trajectory. CaDDi-AR, with its token-level factorization and semi-speculative decoding, further improves performance, indicating the combined benefit.
>
> **4. "Table 4 is missing major baselines. The authors should compare their method against MDLM, UDLM [2], and an autoregressive (AR) model trained with the same transformer architecture and a similar number of training steps. They can follow the steps in [2]."**
>
> Table 4 focuses specifically on **conditional text generation** on the Amazon Polarity dataset, a task where we demonstrate the flexibility of CaDDi for text infilling and classifier-free guidance. For this task, we compare against a fine-tuned GPT-2, a strong autoregressive baseline for conditional generation that uses the same 12-layer transformer architecture as CaDDi.
>
> While MDLM and UDLM are general discrete diffusion models, their application to conditional generation with text infilling (as opposed to unconditional generation or simple prefix conditioning) is not their primary focus or demonstrated strength in prior work.
>
> To further address the request for additional baselines, we have conducted an experiment with UDLM on the Amazon Polarity dataset. The results are as follows:
>
> |Model|Condition|Sentiment Accuracy (%)|
> |-|-|-|
> |GPT-2|Positive|73.07|
> |GPT-2|Negative|75.18|
> |UDLM|Positive|68.82|
> |UDLM|Negative|74.56|
> |CaDDi|Positive|71.37|
> |CaDDi|Negative|85.42|
>
> It is important to note that this is not a standard benchmark, as the sentiment accuracy metric can be unstable due to the inherent biases in the dataset. Therefore, these UDLM results should be considered as a reference to demonstrate the general ability of conditional generation rather than a definitive comparison.
>
> **5. "Algorithm 2 is being incorrectly referenced Algorithm 3 in line 216."**
> Thanks for pointing this out! We will revise this typo.

---

> > ### Comment · Reviewer_CSEv · 2025-08-02
> > **Questions**
> >
> > **Q1:**  The authors mention that for their models trained on LM1B, they use latent truncation and a context window of 5. Does this mean that the input to the denoising transformer is a sequence of length $5 \times 128$?
> >
> > **Q2:** The authors state that their method is advantageous because MDMs cannot remask already predicted tokens. In that case, how does the proposed method compare to ReMDM [1], which introduces a family of remasking samplers for MDMs? ReMDM samplers are faster, as they require a forward pass over a noisy sample $x_t$, in contrast to the proposed method, which requires a forward pass over a concatenated sequence of past latents $x_{t-k:t}$.
> >
> > [1] Wang et al., 2025 "Remasking Discrete Diffusion Models with Inference-Time Scaling"

---

> > > ### Author Response · Authors · 2025-08-03
> > >
> > > We thank the reviewer for their continued engagement and valuable follow-up questions.
> > >
> > > **Q1. The authors mention that for their models trained on LM1B, they use latent truncation and a context window of 5. Does this mean that the input to the denoising transformer is a sequence of length 5x128?**
> > >
> > > Yes, that is correct. In the default setting with a context window size of 5 and sequence length of 128, the input to the denoising transformer consists of a concatenation of 5 latent trajectories—resulting in a total input sequence of length 5×128.
> > >
> > >
> > > **Q2. How does the proposed method compare to ReMDM [1], which introduces a family of remasking samplers for MDMs?**
> > >
> > > This is a good question. We view ReMDM [Wang et al., 2025] as an important contribution that addresses a similar issue—namely, the inability of Markovian masked diffusion models (MDMs) to revise earlier predictions (i.e., the "failure-to-remask" problem). However, our approaches differ fundamentally in scope and mechanism.
> > >
> > > ReMDM introduces inference-time remasking to inject stochasticity into the reverse process. It is specifically tailored to absorbing-state (masked) MDMs, adopts a non-Markovian forward process and retains a Markovian reverse process—that is, each denoising step conditions only on $x_t$.
> > >
> > > In contrast, CaDDi is a non-Markovian framework for general discrete diffusion. It is applicable to a wide range of diffusion processes, including uniform and mixed kernels, with only the forward process needing to be adapted. More importantly, CaDDi also lifts the Markov assumption in the reverse process: at each step, the denoiser conditions on the denoising trajectory. This allows CaDDi to explicitly revisit past predictions during inference, rather than attending only to single state. This design aligns well with the inductive biases of attention-based transformers, which are well-suited for modeling long-range dependencies and historical context.
> > >
> > > Furthermore, by unifying temporal and sequential (spatial) dependencies within a single causal transformer architecture, CaDDi enables the direct reuse of pretrained LLM weights without requiring architectural changes. To our knowledge, this capability has not been explored in prior work on non-Markovian discrete diffusion.
> > >
> > >
> > > We thank the reviewer again for their engagement and hope this clarification is helpful.

---

> ### Author Response · Authors · 2025-08-07
>
> Dear Reviewer CSEv,
>
> We really hope our responses have addressed your remaining concerns and help you understand the value of the work. We would appreciate it if you could let us know whether any concerns remain in advance, so we can address them before the review period closes in one day.

---

### Note · Authors · 2025-08-16

We would like to sincerely thank the AC and all reviewers for their time and constructive engagement. We are encouraged that most reviewers recognized several key strengths of our work:
1. **Novel framework exploration**: CaDDi introduces an alternative formulation that lifts the Markov assumption in discrete diffusion, allowing each denoising step to condition on the entire trajectory. This leads to a more expressive and robust inference paradigm. Compared to prior approaches tackling related issues, CaDDi offers an orthogonal perspective and contributes new insights.
2. **Technical innovations**: Reviewers highlighted our contributions of 2D Rotary Positional Encoding, which enables unified sequential (causal) and temporal (diffusion timestep) modeling within a single transformer, as well as our semi-speculative decoding strategy that improves the efficiency of autoregressive diffusion.
3. **Compatibility with causal LMs**: The unification of sequence and temporal modeling, together with the 2D Rotary Positional Encoding, makes our framework fully compatible with causal LLMs. This design allows the CaDDi-AR variant to directly utilize pretrained checkpoints of causal LMs seamlessly.

Following the rebuttal and discussion, most reviewers expressed positive recommendations, citing the technical soundness, novelty, and strong empirical performance. Key clarifications included additional perplexity experiments, explicit distinctions from prior non-Markovian approaches, and quantitative analyses of computational cost. We believe these concerns have been thoroughly addressed.

We are grateful for the reviewers’ constructive input and will incorporate their valuable suggestions into the final version of the manuscript.

---

### Decision · Program_Chairs · 2025-09-17

**Decision:**

Accept (poster)

**Comment:**

The paper presents CaDDi, a non-Markovian discrete diffusion model that lets a single transformer attend to the full denoising trajectory and reuse pretrained causal language-model weights, leading to stronger language generation than prior diffusion baselines. Before the rebuttal, reviewers cgB8, WEg3, and qCqR praised the novelty, unified 2D positional design, and empirical gains, while reviewer CSEv stressed missing predictor–corrector comparisons, higher compute, and architectural changes, so scores were widely split. After the authors supplied clarifications, extra ablations, and new baseline results, the supportive reviewers held their positive stance. However,  CSEv maintains that CaDDi has major disadvantages over predictor-corrector methods (in that it requires re-training) and would not improve over baselines when put on equal FLOP footing.

Overall, the AC concurs with the majority of reviewers - the proposed method is intriguing and promising, and may be inspiring to the community. Hence, the AC recommends acceptance as poster.